# A mechanism of uncompetitive inhibition of the serotonin transporter

**Shreyas Bhat[1], Ali El-Kasaby[1], Ameya Kasture[2], Danila Boytsov[1], Julian B Reichelt[1], Thomas Hummel[2], Sonja Sucic[1], Christian Pifl[3], Michael Freissmuth[1], Walter Sandtner[1]***

[1]Institute of Pharmacology and the Gaston H. Glock Research Laboratories for Exploratory Drug Development, Center of Physiology and Pharmacology, Medical University of Vienna, Vienna, Austria; [2]Department of Neurobiology, University of Vienna, Vienna, Austria; [3]Center for Brain Research, Medical University of Vienna, Vienna, Austria

**Abstract** The serotonin transporter (SERT/SLC6A4) is arguably the most extensively studied solute carrier (SLC). During its eponymous action – that is, the retrieval of serotonin from the extracellular space – SERT undergoes a conformational cycle. Typical inhibitors (antidepressant drugs and cocaine), partial and full substrates (amphetamines and their derivatives), and atypical inhibitors (ibogaine analogues) bind preferentially to different states in this cycle. This results in competitive or non-competitive transport inhibition. Here, we explored the action of $N$-formyl-1,3-bis (3,4-methylenedioxyphenyl)-prop-2-yl-amine (ECSI#6) on SERT: inhibition of serotonin uptake by ECSI#6 was enhanced with increasing serotonin concentration. Conversely, the $K_M$ for serotonin was lowered by augmenting ECSI#6. ECSI#6 bound with low affinity to the outward-facing state of SERT but with increased affinity to a potassium-bound state. Electrophysiological recordings showed that ECSI#6 preferentially interacted with the inward-facing state. Kinetic modeling recapitulated the experimental data and verified that uncompetitive inhibition arose from preferential binding of ECSI#6 to the $K^+$-bound, inward-facing conformation of SERT. This binding mode predicted a pharmacochaperoning action of ECSI#6, which was confirmed by examining its effect on the folding-deficient mutant SERT-PG[601,602]AA: preincubation of HEK293 cells with ECSI#6 restored export of SERT-PG[601,602]AA from the endoplasmic reticulum and substrate transport. Similarly, in transgenic flies, the administration of ECSI#6 promoted the delivery of SERT-PG[601,602]AA to the presynaptic specialization of serotonergic neurons. To the best of our knowledge, ECSI#6 is the first example of an uncompetitive SLC inhibitor. Pharmacochaperones endowed with the binding mode of ECSI#6 are attractive, because they can rescue misfolded transporters at concentrations, which cause modest transport inhibition.

**\*For correspondence:** walter.sandtner@meduniwien. ac.at

**Competing interest:** The authors declare that no competing interests exist.

## Editor's evaluation

This study presents the important finding of an unusual uncompetitive inhibitor (ECSI#6) of the serotonin transporter SERT that removes the neurotransmitter serotonin from the synaptic cleft. Through careful and comprehensive analysis, the authors convincingly show that the molecule most likely binds to the inward-facing and $K^+$-bound state and that it assists in folding and targeting the transporter. The work will be of interest to those engaged in biophysical analyses of the serotonin transporter, and colleagues developing pharmacological chaperoning strategies for transporters in general.

## Introduction

The transporters for the monoamines serotonin (SERT), dopamine (DAT), and norepinephrine (NET) are members of the solute carrier-6 (SLC6) family (*Kristensen et al., 2011*). The uptake of their eponymous substrates is fueled by the electrochemical gradient of sodium; in addition, DAT and NET can harvest the membrane potential, while SERT utilizes the potassium gradient (*Bhat et al., 2021c*). These driving forces support a remarkable concentrative power, which allows for effective removal of the cognate substrates from the extracellular space. SERT, NET, and DAT are closely related. Accordingly, their pharmacology shows a continuum from compounds, which bind to all three transporters (e.g., cocaine), to drugs with exquisite selectivity (e.g., selective serotonin reuptake inhibitors [SSRIs]). Monoamine transporters are targets for both, therapeutically relevant drugs (e.g., antidepressants) and for illicit substances. Accordingly, the chemical space of ligands has been extensively studied resulting in a very rich pharmacology (*Sitte and Freissmuth, 2015*). Originally, ligands for monoamine transporters have been classified as either non-transportable inhibitors or exogenous substrates/releasers. Inhibitors bind to and block reuptake through the transporter (e.g., cocaine, tricyclic antidepressants, SSRIs). In contrast, exogenous substrates/releasers (e.g., amphetamines) induce efflux of the endogenous monoamine by switching the transporter from the physiological forward transport mode into an exchange mode (*Sitte and Freissmuth, 2015*; *Hasenhuetl et al., 2018*).

More recently, the prevailing dichotomous classification was challenged by the discovery of compounds, which act as partial substrates and/or atypical inhibitors of monoamine transporter (*Rothman et al., 2012*; *Schmitt et al., 2013*; *Sandtner et al., 2016*; *Hasenhuetl et al., 2019*; *Niello et al., 2019*; *Niello et al., 2020*). Partial substrates do not elicit reverse transport to the same extent as full releasers (*Rothman et al., 2012*; *Sandtner et al., 2016*). In fact, there is a continuum, which ranges from full releasers over partial releasers to atypical inhibitors: minor modifications of the amphetamine- or cathinone-based structures suffice to reduce the efficacy of release and to eventually produce atypical inhibitors or allosteric modulators (*Sandtner et al., 2016*; *Niello et al., 2019*). Typical inhibitors bind to the outward-facing conformation of the monoamine transporters. In contrast, atypical inhibitors trap the transporter in distinct conformational states, which are visited during the transport cycle (*Bhat et al., 2019*). Numerous disease-associated mutations have been identified in neurotransmitter transporters, which result in misfolding of the protein (*Bhat et al., 2021b*). Atypical inhibitors and partial substrates are of interest, because they can act as pharmacochaperones and correct the folding defect (*Bhat et al., 2019*; *Bhat et al., 2017*). Here, we studied the inhibition of SERT by the amphetamine derivative ECSI#6 [*N*-formyl-1,3-bis (3,4-methylenedioxyphenyl)-prop-2-yl-amine], which was identified as a synthesis byproduct of 3,4-methylenedioxy-methamphetamine ('ecstasy') (*Pifl et al., 2005*). Our experiments show that ECSI#6 is, to the best of our knowledge, the first reported uncompetitive inhibitor of SERT. This mode of inhibition was accounted for by preferential binding of ECSI#6 to the inward-facing state. Finally, consistent with this binding mode, ECSI#6 also acted as a pharmacochaperone: cellular preincubation with ECSI#6 restored export from the endoplasmic reticulum (ER) and substrate transport by a misfolded SERT variant.

## Results

### Cocaine, noribogaine, and ECSI#6 exhibit differences in the ability to block SERT uptake

Affinity estimates of exogenous ligands to monoamine transporters can be evaluated by their ability to block either the functional uptake of a tritiated substrate or binding of a high-affinity radiolabeled inhibitor to the transporter. For the uptake inhibition experiments, we calculated $IC_{50s}$ for the ability of cocaine, noribogaine, or ECSI#6 to block transport of 0.1 µM [$^3$H]5-HT by SERT either in the absence ($IC_{50(0.1)}$) or prior preincubation of either 1 µM ($IC_{50(1)}$) or 10 µM ($IC_{50(10)}$) cold 5-HT. Cocaine blocks SERT uptake of [$^3$H]5-HT with an $IC_{50(0.1)}$ of 9.96 µM (95% CI: 8.7–11.4, *Figure 1A*, solid blue curve). The $IC_{50(1)}$ was indistinguishable from $IC_{50(0.1)}$ (*Figure 1A*, blue dashed curve, $IC_{50}$=9.8 µM, 95% CI: 7.1–13.6), while $IC_{50(10)}$ was right shifted by ~4-fold (*Figure 1A*, blue dotted curve, $IC_{50}$=34.6 µM, 95% CI: 24.8–48.3). This right shift is expected and readily explained; increasing concentrations of unlabeled 5-HT in the uptake inhibition assay inhibits cocaine binding to SERT in a competitive manner. Noribogaine, on the other hand, blocked substrate uptake of SERT more potently in the presence of unlabeled 5-HT (cf. solid, dashed, and dotted green lines, *Figure 1B*, $IC_{50(0.1)}$: 5.68 µM [95% CI: 4.36–7.39], $IC_{50(1)}$:

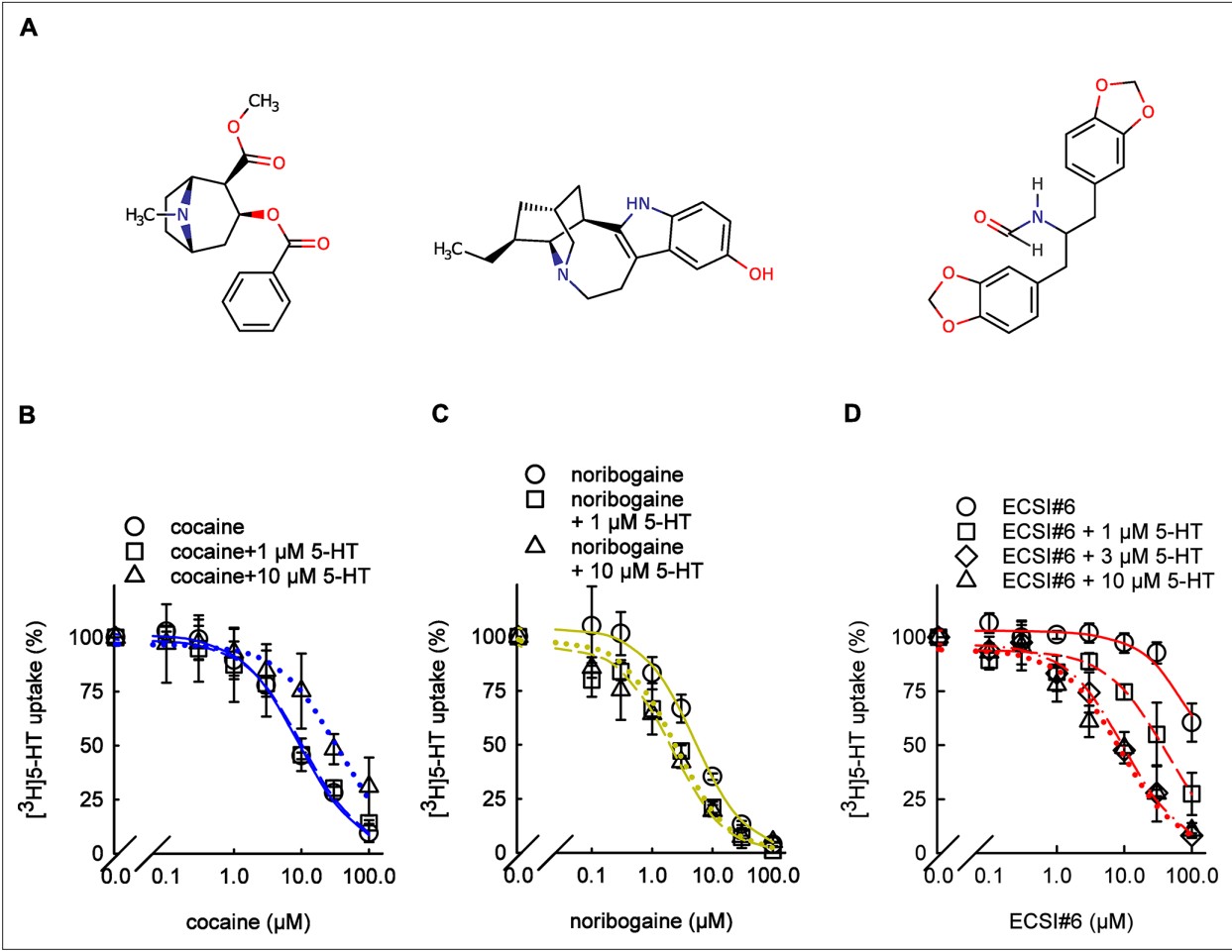

**Figure 1.** Inhibition of serotonin transporter (SERT)-mediated [³H]5-HT uptake by cocaine, noribogaine, and ECSI#6. (**A**) The chemical structures of cocaine, noribogaine, and ECSI#6. (**B–D**) HEK293 cells stably expressing wild-type YFP-SERT (~30,000 per well) were seeded onto 96-well plates. After 24 hr, inhibition of substrate uptake by cocaine (**B**), noribogaine (**C**), or ECSI#6 (**D**) was determined as outlined under Materials and methods in the absence of additional unlabeled 5-HT (circles and solid lines, $IC_{50(0.1)}$) or in the presence of 1 µM (squares and dashed lines, $IC_{50(1)}$), 3 µM (diamonds and dashed-dotted lines in D; $IC_{50(3)}$), or 10 µM 5-HT (triangles, dotted lines, $IC_{50(10)}$) of unlabeled 5-HT. Paroxetine (10 µM) was used to determine nonspecific uptake, which was ~5% of total uptake. Uptake was normalized to the specific uptake in the absence of inhibitors (4.2±0.6 pmol min⁻¹ 10⁻⁶ cells), which was set to 100% (i.e., no inhibition) to account for inter-experimental variations. Data represented are the means ± SD (error bars) from three independent experiments done in triplicate. The curves were generated by fitting a sigmoidal function through data points normalized between 100% and no uptake. The concentrations giving half-maximum inhibition (IC₅₀) were (means and 95% confidence interval in parenthesis): cocaine - $IC_{50(0.1)}$=9.9 µM (8.7–11.4), $IC_{50(1)}$=9.8 µM (7.1–13.5), $IC_{50(10)}$=34.6 µM (24.8–48.3); noribogaine - $IC_{50(0.1)}$=5.7 µM (4.3–7.3), $IC_{50(1)}$=2.4 µM (1.8–3.1), $IC_{50(10)}$=1.9 µM (1.5–2.4); ECSI#6 - $IC_{50(0.1)}$=440.5 µM (325.8–595.6), $IC_{50(1)}$=74.1 µM (55.4–99.2), $IC_{50(3)}$=25.6 µM (19.1–34.2), $IC_{50(10)}$=19.3 µM (13.2–28.1).

2.39 µM [95% CI: 1.84–3.09], $IC_{50(10)}$: 1.89 µM [95% CI: 1.46–2.44]). In the presence of unlabeled 5-HT, this left shift in potency was more prominent with ECSI#6 (cf. solid, dashed, and dotted red lines, *Figure 1B*, $IC_{50(0.1)}$: 440.5 µM [95% CI: 325.8–595.6], $IC_{50(1)}$: 74.14 µM [95% CI: 55.4–99.22], $IC_{50(10)}$: 19.28 µM [95% CI: 13.23–28.09]). These observations clearly show that noribogaine and ECSI#6 bind to SERT with higher affinity in the presence of a substrate. Thus, their mode of binding to the transporter differs from that of cocaine.

## Cocaine, noribogaine, and ECSI#6 bind to different SERT conformational intermediates

The data summarized in *Figure 1* indicate that ECSI#6 and noribogaine differ in their mode of binding from cocaine. There are two explanations for the left shift in the uptake inhibition by ECSI#6 (and by noribogaine) in the presence of increasing concentrations of 5-HT: (i) binding of 5-HT to SERT results in a conformational transition, which exposes a binding pocket for ECSI#6 (and for noribogaine). (ii)

Alternatively, uptake of 5-HT by SERT results in the accumulation of the $K^+$-bound conformational intermediate(s), which display a higher affinity for ECSI#6 or noribogaine (we emphasize that accumulation in the $K^+$-bound intermediates is a consequence of the slow return of the $K^+$-bound transporters from the inward- to the outward-facing conformation). The first model posits that serotonin and ECSI#6 are bound simultaneously. This has been observed with serotonin and carbamazepine, which can bind concomitantly to the outward-facing state of SERT (*Jacobs et al., 2007*). We resorted to radioligand binding experiments with [³H]citalopram to address which of the two possibilities can explain the experimental observation: [³H]citalopram is a radiolabeled high-affinity inhibitor of SERT, which binds to the outward-facing conformation. Membranes harboring SERT were incubated with 3 nM [³H]citalopram and logarithmically spaced concentrations of ECSI#6 (and the reference compounds cocaine and noribogaine) in the presence of 120 mM NaCl, which promotes the outward-facing conformation. It is evident from *Figure 2C* that addition of 1 or 10 µM 5-HT shifted the displacement curve of ECSI#6 to the right to an extent which was comparable to that seen with cocaine (*Figure 2A*) and noribogaine (*Figure 2B*). We transformed the data summarized in *Figure 2A–C* by plotting the reciprocal of bound radioligand as a function of inhibitor concentration to yield Dixon plots (*Figure 2D–F*): the x-intercept corresponds to -$IC_{50}$ of the inhibitor (*Bulling et al., 2012*). Thus, Dixon plots allow for differentiating mutually exclusive from mutually non-exclusive binding, if one inhibitor (i.e., cocaine, noribogaine, or ECSI#6) is examined at a fixed concentration of the second inhibitor (i.e., serotonin) (*Segel, 1975*): if binding of the two inhibitors is mutually non-exclusive, a family of lines of progressively increasing slope, which intersect at -$IC_{50}$, is to be seen. In contrast, if the two inhibitors bind to the same site, the slope of the inhibition curves is not affected and the x-intercept (i.e., -$IC_{50}$ of the variable inhibitor) is shifted to more negative values. It is evident from *Figure 2D–E* that the presence of 1 and 10 µM serotonin progressively shifted the (expected) x-intercept for cocaine (*Figure 2D*), noribogaine (*Figure 2E*), and ECSI#6 (*Figure 2D*). Thus, binding of serotonin and one of these three ligands to SERT was mutually exclusive.

We also explored the alternative hypothesis, that is, that ECSI#6 bound preferentially to a $K^+$-bound state of SERT. $K^+$ promotes the return step of SERT and thus drives SERT into the outward-facing state. This allows for measuring detectable levels of radioligand binding in the presence of 120 mM $K^+$ (*Bhat et al., 2021c*; *Schicker et al., 2012*). We compared the ability of ECSI#6 and of the reference compounds cocaine and noribogaine to displace [³H]citalopram in the presence of 120 mM NaCl and 120 mM KCl. As expected, cocaine was substantially less potent (by an order of magnitude) in inhibiting [³H]citalopram binding in the presence of 120 mM KCl than in the presence of 120 mM NaCl (*Figure 2D*). This indicates that cocaine binds more readily to $Na^+$-bound SERT. This distinct binding preference was not obvious with noribogaine (*Figure 2E*). Importantly and in contrast to cocaine, ECSI#6 was about 10-fold more potent in displacing [³H]citalopram binding in the presence of 120 mM KCl than in the presence of 120 mM NaCl (*Figure 2E*). Thus ECSI#6 clearly preferred $K^+$-bound SERT.

## Cocaine, noribogaine, and ECSI#6 differ in their mode of SERT inhibition

Cocaine and noribogaine are competitive and non-competitive inhibitors of substrate uptake by SERT, respectively (*Jacobs et al., 2007*; *Bulling et al., 2012*). The mode of inhibition by ECSI#6 was anticipated to differ from either compound because of the pronounced left shift in both, the uptake inhibition curve of ECSI#6 in the presence of increasing serotonin (*Figure 1D*) and the radioligand displacement curve in the presence of high $K^+$ (*Figure 2F*). This was the case: increasing concentrations of ECSI#6 shifted the saturation hyperbolae for 5-HT uptake (*Figure 3C*, top) and caused a reduction in both, $K_M$ (*Figure 3C*, middle panel) and $V_{MAX}$ (*Figure 3C*, bottom panel). This is the hallmark of uncompetitive inhibition. In parallel experiments, we recapitulated competitive and non-competitive inhibition of substrate uptake by cocaine (*Figure 3A*) and by noribogaine (*Figure 3B*), respectively.

## Kinetics of current inhibition by ECSI#6

The different mode of inhibition, which was observed for ECSI#6, implies that it affects one or several partial reactions of the transport cycle in a manner distinct from cocaine and noribogaine. The transport cycle of SERT can be addressed by whole-cell patch clamp recordings; electrophysiological recordings

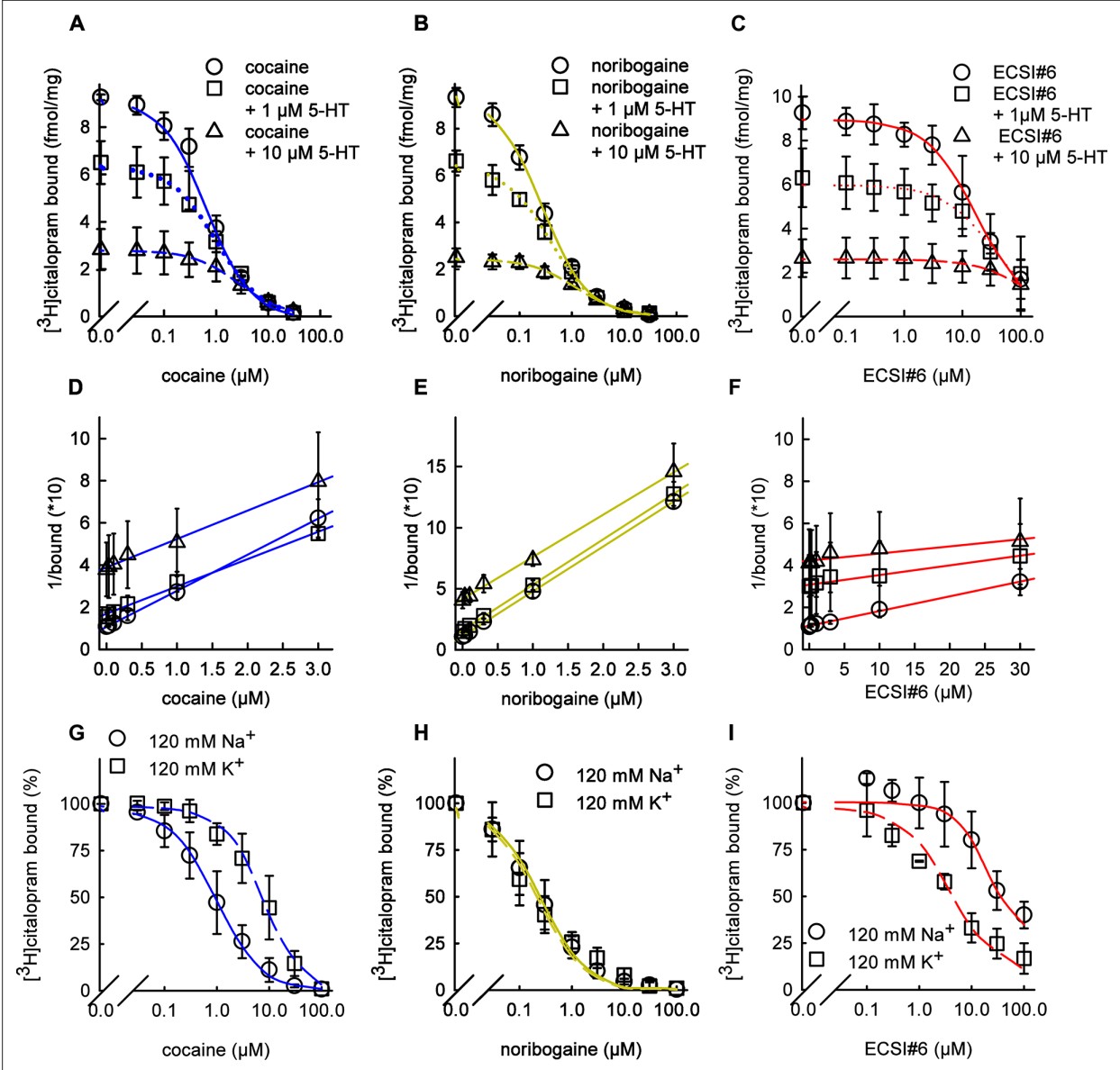

**Figure 2.** Inhibition of [³H]citalopram binding by cocaine, noribogaine, and ECSI#6. (A–G) Inhibition of [³H]citalopram binding to serotonin transporter (SERT) by cocaine (A, D; blue curves), noribogaine (B, E; green curves), or ECSI#6 (C, F; red curves). The binding inhibition assays in (A–C) were performed in the absence (open circles, solid lines) and presence of 1 μM (squares, dashed lines) or of 10 μM 5-HT (triangles, dotted lines) in buffer containing 120 mM Na⁺ with 2–3 μg membranes (0 and 1 μM 5-HT) or 7–8 μg membranes (10 μM 5-HT). (D–F) The data in A–C were transformed to yield Dixon plots. The binding inhibition assays in (G–I) were performed in a binding buffer that contained either 120 mM Na⁺ (circles, solid curves) or 120 mM K⁺ (squares, dashed curves). Assays in the presence of 120 mM NaCl were done with 2–3 μg membranes; for assays with 120 mM KCl, 7–8 μg of membranes were used to improve the dynamic binding range. The specific binding values for assays with the 120 mM NaCl and 120 mM KCl buffer were 25±14 and 21±5 fmol assay⁻¹, respectively. These values were normalized to 100% to account for inter-experiment variability. All data are the means from three independent experiments done in duplicate; error bars represent SD. The curves generated were generated by fitting the data points to the equation for a monophasic inhibition. The concentrations giving half-maximum inhibition (IC$_{50}$ values) were calculated from each plot yielding (95% confidence intervals in parenthesis): (A) cocaine - IC$_{50(\text{no 5-HT})}$=0.66 μM (0.54–0.82), IC$_{50(1\ \mu M\ 5\text{-HT})}$=0.94 μM (0.76–1.09), IC$_{50(10\ \mu M\ 5\text{-HT})}$=2.48 μM (2.08–2.90); (B) noribogaine - IC$_{50(\text{no 5-HT})}$=0.28 μM (0.24–0.31), IC$_{50(1\ \mu M\ 5\text{-HT})}$=0.32 μM (0.26–0.42), IC$_{50(10\ \mu M\ 5\text{-HT})}$=1.10 μM (0.92–1.31); (C) ECSI#6 - IC$_{50(\text{no 5-HT})}$=16.02 μM (10.96–22.42), IC$_{50(1\mu M\ 5\text{-HT})}$=28.6 μM (20.2–40.0), IC$_{50(10\ \mu M\ 5\text{-HT})}$=60.0 μM (38.2–94.2); (D) cocaine - IC$_{50(120\ Na^+)}$=0.82 μM (0.62–1.10), IC$_{50(120\ K^+)}$=7.94 μM (5.34–9.01); (E) noribogaine - IC$_{50(120\ Na^+)}$=0.24 μM (0.18–0.32), IC$_{50(120\ K^+)}$=0.20 μM (0.16–0.24); (F) ECSI#6 - IC$_{50(120\ Na^+)}$=46.2 μM (31.6–67.6), IC$_{50(120\ K^+)}$=4.18 μM (2.58–5.88). Paroxetine (10 μM) was used to determine nonspecific binding, which was ≤10% of total binding.

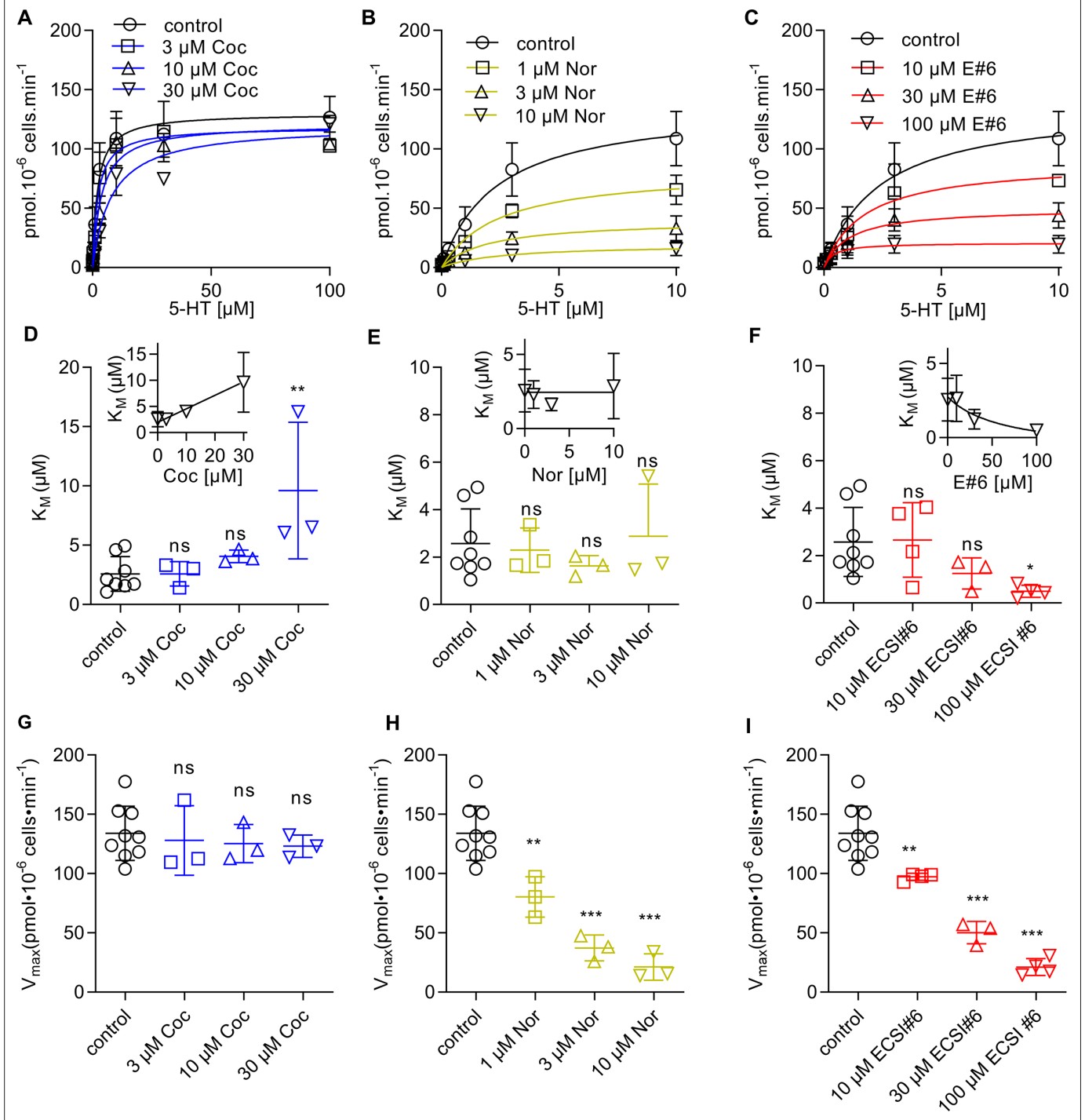

**Figure 3.** Different modes of inhibition by cocaine, noribogaine, and ECSI#6 of substrate uptake by serotonin transporter (SERT). HEK293 cells stably expressing YFP-SERT (30,000/well) were preincubated in buffer in the absence (control, black circles) or presence of the indicated concentrations of cocaine (**A**), noribogaine (**B**), or ECSI#6 (C), for 10 min; subsequently, the uptake reaction was initiated as outlined in Materials and methods. In the absence of any inhibitor (black circles and black lines, control), the $K_M$ and $V_{MAX}$ of 5-HT transport by SERT was 2.3 µM (95% CI, 1.5–3.2) and 137.3 pmol min$^{-1}$ 10$^{-6}$ cells (95% CI, 117.4–157.3); these control curves are identical in all panels. (**A**) In the presence of 3 µM (blue squares), 10 µM (blue upward triangles), and 30 µM cocaine (blue downward triangles), there was a progressive increase in $K_M$-values (panel **D**) with $K_M$ = 2.2 µM [95% CI, 1.1–3.3], 3.7 µM [95% CI, 1.9–5.5], and 8.1 µM [95% CI, 1.9–5.5], respectively. The inset in panel D shows the same data on a linear abscissa to visualize the linear relation between apparent $K_M$ and cocaine concentration. In contrast to the $K_M$ values, the $V_{max}$-values remained constant (panel **G**, $V_{MAX}$ = 118.3 pmol min$^{-1}$ 10$^{-6}$ cells [95% CI, 117.4–157.3], 121.1 pmol min$^{-1}$ 10$^{-6}$ cells [95% CI, 105.5–136.8], and 120.1 pmol min$^{-1}$ 10$^{-6}$ cells [95% CI, 104.6–135.6], respectively). (**B**) Preincubation with 1 µM (yellow squares), 3 µM (yellow upward triangles), and 10 µM noribogaine (yellow downward triangles) did not change the

*Figure 3 continued on next page*

*Figure 3 continued*

$K_M$-values (panel **E** with $K_M$ = 2.4 µM [95% CI, 1.4–3.5], $K_M$ = 1.9 µM [95% CI, 0.6–3.3], and 2.4 µM [95% CI, 0.6–5.5], respectively). The inset shows the same data on a linear abscissa. The $V_{MAX}$ on the other hand was reduced in a concentration-dependent manner (panel **H** with $V_{MAX}$ = 82.4 pmol min$^{-1}$ 10$^{-6}$ cells [95% CI, 69.3–95.4], 39.9 pmol min$^{-1}$ 10$^{-6}$ cells [95% CI, 30.6–49.2], and 19.5 pmol min$^{-1}$ 10$^{-6}$ cells [95% CI, 13.9–24.9], respectively). (**C**) Preincubation with 10 µM (red squares), 30 µM (red upward triangles), and 100 µM ECSI#6 (red downward triangles) led to a drop in both, the $K_M$-values (panel **F**, $K_M$ = 1.7 µM [95% CI, 0.5–2.9], 1.1 µM [95% CI, 0.2–1.9], and 0.4 µM [95% CI, 0.0–0.8], respectively; the inset shows the same data on a linear abscissa) and the $V_{MAX}$-values (panel **I**, $V_{MAX}$ = 89.5 pmol min$^{-1}$ 10$^{-6}$ cells [95% CI, 68.5–110.0], 49.9 pmol min$^{-1}$ 10$^{-6}$ cells [95% CI, 37.5–62.3], and 20.8 pmol min$^{-1}$ 10$^{-6}$ cells [95% CI, 15.6–26.0], respectively). Data are the means ± SD (error bars) from at least three independent experiments done in triplicate. The curves were generated by fitting the data points to the equation for a rectangular hyperbola. Control $K_M$- and $V_{MAX}$-values were pooled and are hence the same in panels **D–F** and **G–I**, respectively. Statistical comparisons were done by one-way ANOVA followed by Dunnett's post hoc test to verify significant differences vs. control (*p<0.05, **p<0.01, ***p<0.001).

of currents through SERT provide unmatched temporal resolution of ligand and co-substrate binding and unbinding events (*Hasenhuetl et al., 2016*; *Hasenhuetl et al., 2015*). When challenged with a substrate like 5-HT or amphetamines, SERT carries inward currents. These are comprised of an initial transient peak current and a steady current. The peak current reflects the initial synchronized movement of substrate and co-substrate through the membrane electric field. The steady current arises from an uncoupled current; the underlying channel mode is visited from the K$^+$-bound inward-facing state. Thus, the steady current reflects the continuous cycling of the transporter in the forward transport mode (*Schicker et al., 2012*; *Sandtner et al., 2014*). In contrast, inhibitors only elicit a minor peak current on application to SERT-expressing cells (*Burtscher et al., 2018*). As shown in *Figure 4A*, cocaine and noribogaine induced transient peak currents, while ECSI#6 failed to do so. In the presence of physiological ion gradients and in the absence of substrate, SERT dwells primarily in the outward-facing conformation. A precondition for a ligand to elicit a peak current is, therefore, the ability to bind to outward-facing states. The failure of ECSI#6 to produce a peak current is, thus, consistent with the interpretation that this compound binds exclusively to inward-facing states. It should be noted, however, that cocaine and noribogaine carry a charge while ECSI#6 does not. Thus, it is not possible to formally rule out the alternative explanation that ECSI#6 fails to generate a peak current, because it is uncharged. We relied on measuring the time course of inhibition of the substrate-induced steady current through SERT to infer the kinetics of inhibitor binding: we first applied 5-HT at a saturating concentration, which elicited the peak current and the steady-state current (*Figure 4B*). After 5 s, the cells were superfused with inhibitors in the continuous presence of the substrate. ECSI#6 and the reference compounds cocaine and noribogaine (red, blue, and green trace, respectively, in the magnified section in *Figure 4B*) led to a rapid inhibition of the substrate-induced current, which was rapidly reversed by removal of the inhibitor. The inhibitor-induced suppression of the steady current was adequately described by a mono-exponential decay. The rate constant reflects the apparent rate of association ($k_{app}$) of the inhibitors. In a simple bimolecular reaction (corresponding to binding of the inhibitor to SERT), raising the inhibitor concentration is predicted to result in accelerated binding. This was not the case for cocaine (*Figure 4C*): as shown previously (*Bulling et al., 2012*), the $k_{app}$ of cocaine only increased in the very low concentration range and leveled off at about 2 s$^{-1}$. This is expected for a typical inhibitor, which interacts with the outward-facing conformation of SERT, because the binding is limited by the return step from the inward-facing to the outward-facing state. This is the rate-limiting reaction in the transport cycle of SERT (*Schicker et al., 2012*; *Hasenhuetl et al., 2015*). In contrast, $k_{app}$ for both, noribogaine (*Figure 4D*) and ECSI#6 (*Figure 4E*), increased in a linear manner as their concentrations were raised. This can only be achieved, if noribogaine and ECSI#6 bind to the inward-facing conformation of SERT directly. Noribogaine was ~10-fold more potent than ECSI#6 in imposing the block on the substrate-induced current through SERT. We obtained estimates for the $k_{on}$ and $k_{off}$ of noribogaine and ECSI#6, respectively, from the slopes and the y-intercepts of the linear regression in panels D and E. The extracted $k_{on}$ and $k_{off}$ values were 1.7*10$^5$±1.6*10$^4$ M$^{-1}$ s$^{-1}$ and 1.4±0.3 s$^{-1}$ and 2.1*10$^4$±3.5*10$^3$ M$^{-1}$ s$^{-1}$ and 1.7±0.5 s$^{-1}$, for noribogaine and for ECSI#6, respectively. However, these rate constants do not allow for estimating a kinetically calculated $K_D$. In fact, the rate of the inhibitor-induced current block is comprised of both (i) the binding rates of the inhibitor and (ii) the rates of the conformational transitions, which SERT undergoes during substrate transport. Thus, the protocol employed (*Figure 4B*) does not afford an accurate determination of inhibitor binding rates.

The protocol depicted in *Figure 4B* can also be used to gauge the apparent affinity of ECSI#6 for SERT in the presence of 5-HT. Plotted in *Figure 4G* is the block of the serotonin-induced current as

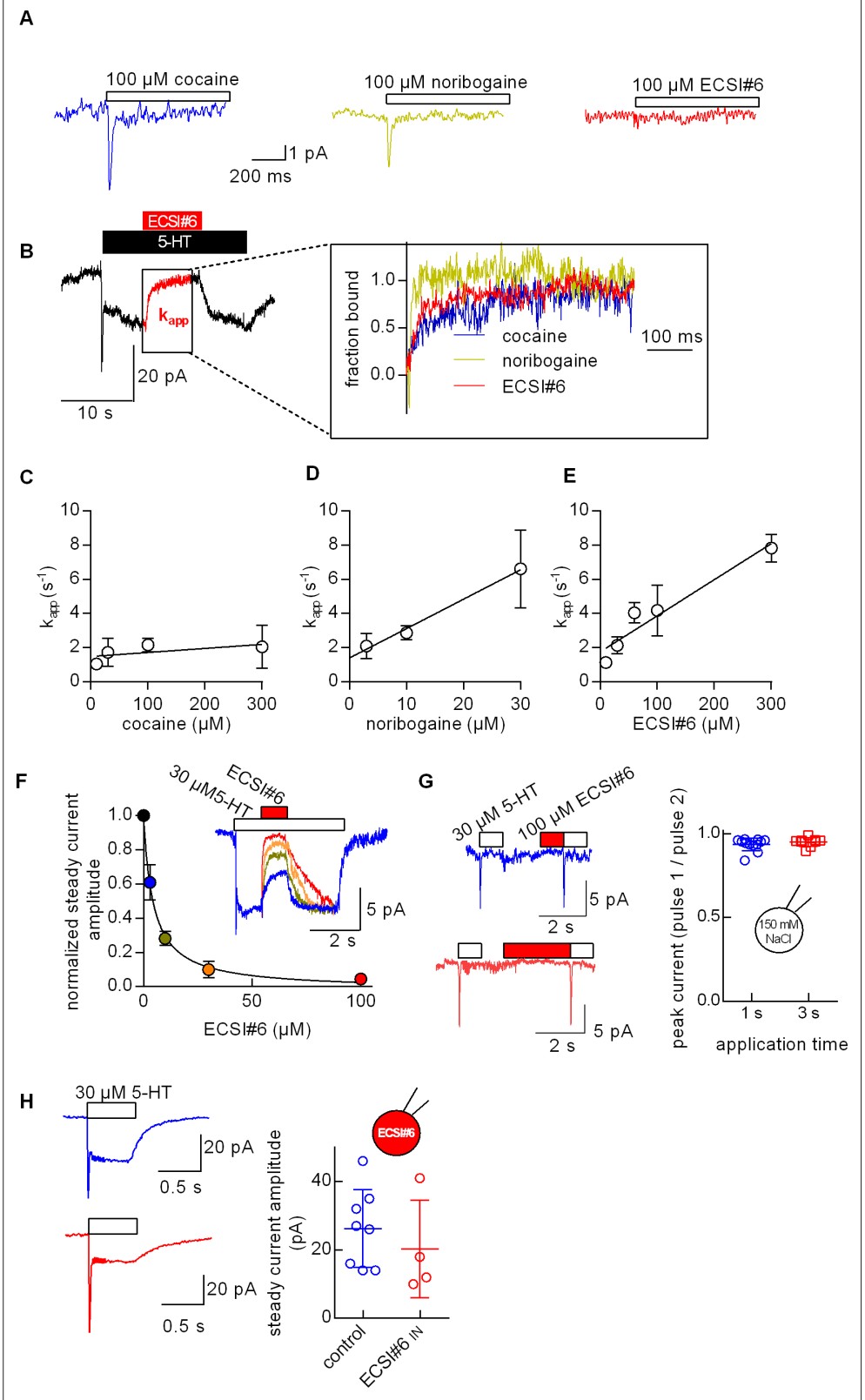

**Figure 4.** Kinetics of current inhibition by cocaine, noribogaine, and ECSI#6. Single HEK293 cells stably expressing GFP-serotonin transporter (SERT) were voltage-clamped to –60 mV using the whole-cell patch clamp technique under physiological ionic gradients. (**A**) Representative recording of a cell superfused with cocaine (100 μM, left-hand trace), noribogaine (100 μM, middle trace), or ECSI#6 (100 μM, right-hand trace). (**B**) Cells were initially

*Figure 4 continued on next page*

*Figure 4 continued*

challenged with 10 µM 5-HT. After 5 s, the cells were superfused with varying concentrations of either cocaine, noribogaine, or ECSI#6 for 5 s in the continuous presence of 10 µM 5-HT, which resulted in current inhibition. Thereafter, the compounds were removed by superfusion with 10 µM 5-HT alone for 10 s to monitor the recovery of the steady current. The left-hand panel shows the original trace of a representative current recorded after sequential superfusion with 10 µM 5-HT for 5 s, which triggered a peak current followed by a steady current, 100 µM ECSI#6 in the continuous presence of 10 µM 5-HT, which led to complete suppression of the current (highlighted in red), and 10 µM 5-HT, which resulted in the recovery of the steady current. The right-hand panel shows magnified representative segments of the traces recorded from different cells, which had been superfused with either cocaine (blue, 100 µM), noribogaine (green, 30 µM), or ECSI#6 (red, 100 µM). The time course of current inhibition was fitted to the equation for a mono-exponential decay to estimate the apparent rate constant ($k_{app}$). (**C, D, E**) Analysis of the kinetics of current inhibition by cocaine (open circles), noribogaine (open triangles), and ECSI#6 (open squares), respectively. The $k_{app}$-values were derived from experiments done as in panel A with the indicated concentrations of the compounds. Data are means ± SD ≥5 independent recordings. The $k_{app}$-values were plotted against the concentration. The slopes of the resulting lines were calculated by linear regression. The slope for cocaine did not deviate in a statistically significant manner from zero (F-test; p=0.12), while those for noribogaine and ECSI#6 did (p<0.0001 in both instances). (**F**) Assessment of the apparent ECSI#6 affinity for SERT in the presence of 5-HT. The inset in panel G shows representative current traces obtained from a cell stably expressing SERT. The current was evoked by rapid application of 30 µM 5-HT to the cell. Once the current had reached the steady level, increasing concentrations of ECSI#6 were applied in the continuous presence of 5-HT for a period of 0.5 s (inset). Application of ECSI#6 led to a concentration-dependent reduction in the current amplitude. The graph shows the fraction of unblocked current as a function of the ECSI#6 concentration. These data were fitted to an inhibition curve. The $IC_{50}$ value estimated by the fit was 4.5±0.5 µM (n=5). (**G**) Assessment of the apparent affinity of ECSI#6 for SERT in the absence of 5-HT: the peak current was isolated by employing an internal solution containing 150 mM NaCl (which suppressed the steady current). The current traces show 5-HT-induced peak currents before and after pre-application of 100 µM ECSI#6 for 1 s (upper panel in blue) and 3 s (lower panel in red): pre-application of 100 µM ECSI#6 did not result in the reduction of the peak current amplitude consistent with a very low affinity for SERT in the absence of 5-HT. The panel on the right shows the result of independent measurements (n=12, prepulse 1 s; n=9, prepulse 3 s). Plotted is the ratio of the amplitude of peak 2 (i.e., after ECSI# pre-application) and peak 1 (i.e., control). (**H**) Effect of intracellular application of ECSI#6 on 5-HT-induced currents: the left-hand panel shows representative currents elicited by the application of 30 µM 5-HT in the absence (upper traces in blue) and presence of 100 µM intracellular ECSI#6 (lower traces in red). The right-hand panel summarizes the current amplitudes obtained from individual cells measured in the absence (blue circles) and presence of ECSI#6 (red circles). When applied from the intracellular side, ECSI#6 only had a modest effect, if any, on the current amplitude.

---

a function of the co-applied ECSI#6 concentration. The current was evoked by a saturating concentration of 5-HT (30 µM) and inhibited by 3, 10, 30, and 100 µM co-applied ECSI#6, respectively (the inset in *Figure 4G* shows representative current traces). A fit of an inhibition curve to the data points yielded an $IC_{50}$ value of approx. 5 µM. This value was lower but still in reasonable agreement, with the $IC_{50}$ obtained in the radioligand uptake assay for the condition where the 5-HT concentration had been saturating (cf. dashed line in *Figure 1C*; 10 µM 5-HT). In the uptake assay the $IC_{50}$ value of ECSI#6 dropped to about 0.5 mM, in the presence of a low 5-HT concentration (i.e., 0.1 µM). In contrast to uptake experiments, electrophysiological recordings also allow for assessing the apparent affinity of ECSI#6 for SERT in the absence of the substrate. This can be achieved by employing the protocol depicted in *Figure 4H* (see representative current traces on the left-hand side): we first applied 30 µM 5-HT to a cell expressing SERT for 0.5 s to elicit a peak current (i.e., a control pulse). We then reapplied 30 µM 5-HT after superfusing the cell with 100 µM ECSI#6 for 1 s (second upper trace in panel H). We chose this time period because it had been sufficient to allow for a full current block in the other protocol (see *Figure 4G*): the amplitude of the peak current following the pre-application of 100 µM ECSI#6 was essentially identical to the prior control pulse. When we pre-applied 100 µM ECSI#6 for a longer period (i.e., 3 s) the amplitude of the two peak currents also remained the same (cf. lower traces in panel H). The right-hand panel shows the summary of several experiments. Plotted in the graph is the ratio of the second and first pulse, which was always close to one. We previously used this protocol to assess the binding kinetics of cocaine, methylphenidate, and desipramine on SERT and DAT. Pre-application of these inhibitors consistently led to a concentration-dependent reduction in the peak current amplitude of the second pulse in comparison to the first (*Hasenhuetl et al., 2015*). The lack of inhibition, thus, indicates that the affinity of ECSI#6 in the absence of 5-HT is

low. To obtain estimates for the affinity of ECSI#6 for SERT in the absence of 5-HT we would need to apply this compound at much higher concentrations. This, however, is not possible, because ECSI#6 is poorly soluble in aqueous solutions (i.e., max. 0.03 mg mL$^{-1}$) (*chemicalize, 2022*).

*Figure 4I* shows representative traces of 5-HT-induced currents recorded from SERT-expressing cells in the absence (in blue) and presence of 100 µM ECSI#6 (in red) in the electrode solution: when applied from the intracellular side, ECSI#6 did not cause an appreciable current block. The right-hand panel summarizes the current amplitude obtained from cells measured in the absence (blue open circles) and the presence of intracellular ECSI#6 (open circles in red). These data seem to indicate that ECSI#6 binds to SERT from the extracellular side. Yet this conclusion can be challenged based on the following consideration: in earlier experiments, ibogaine, the parent compound of noribogaine, was found to block HERG channels when applied from the bath solution but failed to do so when added to the electrode solution (*Thurner et al., 2014*). However, at a lower intracellular pH (i.e., pH 5.5), ibogaine gained the ability to inhibit HERG from the intracellular side (i.e., via application through the electrode). Conversely, ibogaine was less effective when applied to an acidified bath solution. These observations led to the conclusion that ibogaine blocked HERG from the cytosolic side: because the molecule in its neutral form was so diffusive, a low intracellular pH was required to force its protonation and thus preclude diffusion from the interior of the cell. ECSI#6 is presumed to also be very diffusible given its estimated log p value and polar surface area of 2.48 and 66 Å$^2$, respectively (*chemicalize, 2022*). However, ECSI#6 harbors an amide nitrogen (see *Figure 1A*) and thus remains neutral in the experimentally accessible pH range. Hence, it is not possible to verify to which side of SERT it binds.

## ECSI#6 can rescue a folding-deficient SERT mutant

In the ER, the folding trajectory of SERT moves through the inward-facing conformation (*Chiba et al., 2014*; *Freissmuth et al., 2017*). Compounds, which bind to the inward-facing conformation, act as pharmacochaperones: they rescue folding-deficient mutants of SERT and of DAT (*Bhat et al., 2019*). Taken together, the experiments summarized in *Figures 1–4* highlight that ECSI#6 binds to SERT in a unique mode, which favors the K$^+$-bound inward-facing state. Thus, ECSI#6 was predicted to have pharmacochaperoning activity. This prediction was verified by using SERT-PG$^{601,602}$AA. This mutant has a severe folding defect, the bulk of the protein is trapped in the ER at an early stage of the SERT folding trajectory (*El-Kasaby et al., 2010*; *El-Kasaby et al., 2014*). Accordingly, only a small fraction of SERT-PG$^{601,602}$AA reaches the cell surface, where it supports substrate uptake, but the folding defect can be corrected, if cells are preincubated with noribogaine (*Bhat et al., 2021a*): as shown in *Figure 5A*, in paired transient transfections, in HEK293 cells expressing SERT-PG$^{601,602}$AA, substrate uptake was only ~10% of cells expressing wild-type SERT. If HEK293 cells were preincubated with 30 µM of cocaine, noribogaine or ECSI#6, appreciable functional rescue was observed with ECSI#6 and the reference compound noribogaine; cocaine was ineffective (*Figure 5B*). This pharmacochaperoning action of ECSI#6 and of noribogaine was concentration-dependent and saturable (*Figure 5C*): noribogaine was more potent (EC$_{50}$=0.46 µM) and more efficacious (E$_{max}$ = 69.5% of the transport velocity observed in cells expressing wild-type SERT) than ECSI#6 (EC$_{50}$=2.9 µM; E$_{max}$ = 52%).

We independently confirmed that the rescue of SERT-PG$^{601,602}$AA by ECSI#6 (and the reference compound noribogaine) was indeed due to increased ER export of the mutant by determining the glycosylation state of the protein. Membrane proteins acquire N-linked core glycans co-translationally in the ER; in the Golgi apparatus, they incorporate additional sugar moieties to achieve mature glycosylation. Two species were visualized by immunoblotting of whole-cell lysates from cells transiently expressing wild-type SERT (lane 1 in representative blot displayed in *Figure 5D*): (i) the band at approximately 75 kDa (labeled C) corresponds to the ER-resident core-glycosylated species (*El-Kasaby et al., 2010*; *Bhat et al., 2021a*); (ii) a broad smear migrating in the range of 90–110 kDa (labeled M) represents the transporters carrying mature glycans (*Freissmuth et al., 2017*). Lysates from cells expressing SERT-PG$^{601,602}$AA alone show the predominant presence of the core-glycosylated species indicating ER retention (lane 2, *Figure 5D*). If lysates from cells expressing SERT-PG$^{601,602}$AA were treated for 24 hr with noribogaine (lanes 4–8 in *Figure 5D*) or with ECSI#6 (lanes 9–13 in *Figure 5D*), there was a concentration-dependent increase in the appearance of the mature glycosylated species. Noribogaine was more potent than ECSI#6 (*Figure 5E*). We note that the EC$_{50}$-values for enhancing the accumulation of the mature glycosylated SERT (*Figure 5E*) differed from those for restoring uptake. This presumably reflects the modest precision, which band densities can be

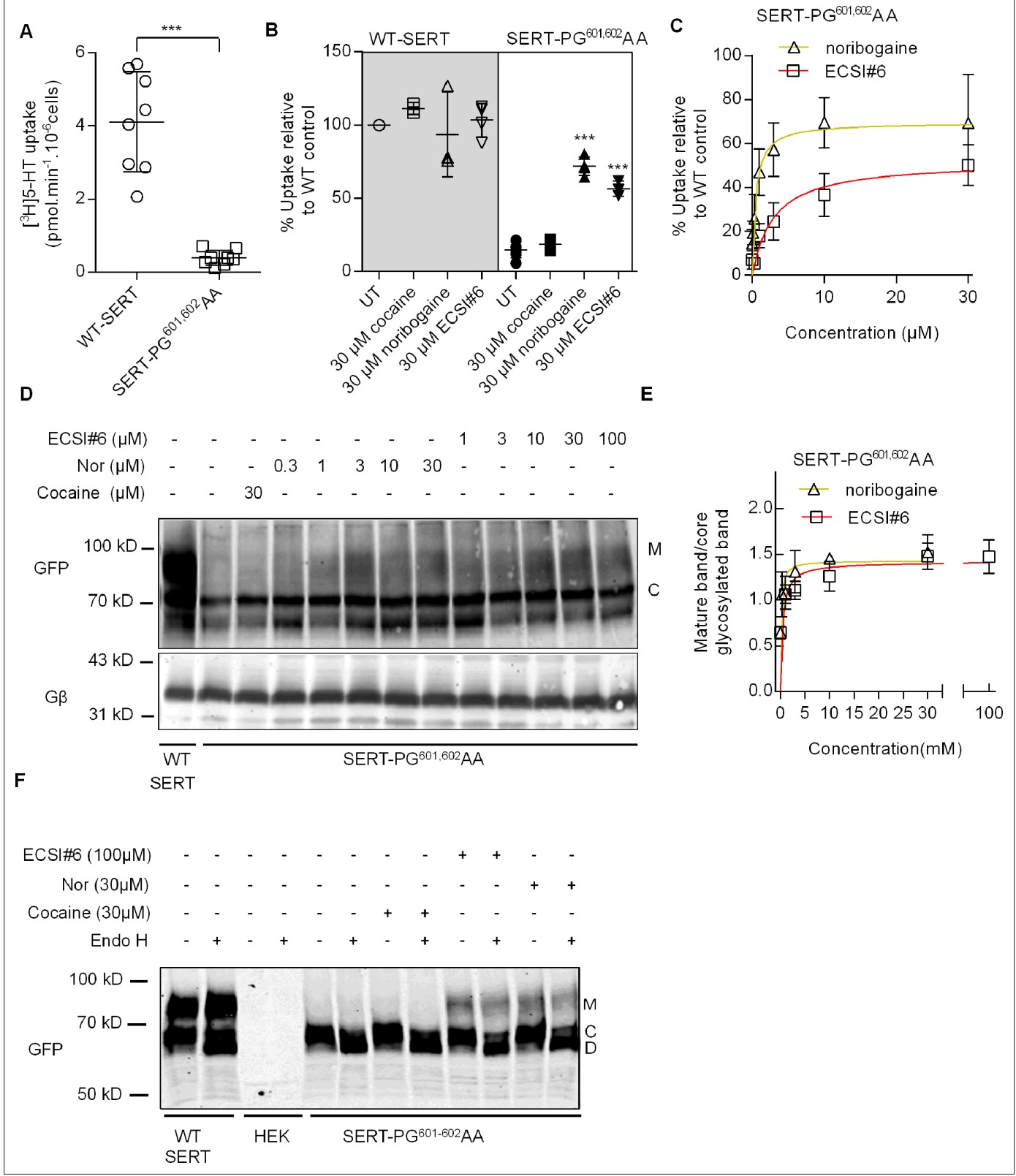

**Figure 5.** ECSI#6 can rescue folding-deficient SERT-PG601-602AA. [³H]5-HT uptake by and glycosylation pattern of hSERT-PG$^{601,602}$AA expressed in HEK293 cells incubated with cocaine, noribogaine, or ECSI#6. (**A**) Comparison of [³H]5-HT uptake by HEK293 cells transiently expressing either wild-type serotonin transporter (WT SERT) or SERT-PG$^{601,602}$AA. Cellular uptake of the substrate was measured with 0.1 μM[³H]5-HT as outlined under Materials and methods and amounted to (means ± SD) 4.1±1.3 pmol min⁻¹ 10⁻⁶ cells and 0.39±0.20 pmol min⁻¹ 10⁻⁶ cells for wild-type SERT and SERT-

*Figure 5 continued on next page*

*Figure 5 continued*

PG$^{601,602}$AA, respectively. Each symbol represents the result of an individual experiment (done in triplicate). The statistical comparison was done by a Mann-Whitney test (p=0.0009). (**B**) Cells transiently expressing either WT-SERT or SERT-PG$^{601,602}$AA were incubated in the presence of 30 µM of either cocaine, noribogaine, or ECSI#6. After 24 hr, cellular substrate uptake was determined with 0.1 µM [$^{3}$H]5-HT. Uptake values from individual conditions were normalized to the ones from untreated cells expressing wild-type SERT (set to 100%) to account for inter-experimental variations. Values from individual experiments (done in triplicate), represented collectively as a box plot, are shown as mean ± SD as follows: WT-SERT untreated (4.1±1.3 pmol min$^{-1}$ 10$^{-6}$ cells set to 100%), WT-SERT+30 µM cocaine (111.4 ± 4.2%), WT-SERT+30 µM noribogaine (93.5 ± 28.8%), WT-SERT+30 µM ECSI#6 (103.3 ± 11.3%), SERT-PG$^{601,602}$AA untreated (14.8 ± 5.3%), SERT-PG$^{601,602}$AA+30 µM cocaine (18.6 ± 3.9%), SERT-PG$^{601,602}$AA+30 µM noribogaine (71.9 ± 6.4%), SERT-PG$^{601,602}$AA+30 µM ECSI#6 (56.5 ± 5.1%). The statistical comparison of untreated cells expressing SERT-PG$^{601,602}$AA and their treated counterparts was done by one-way ANOVA followed by post hoc Dunnett's multiple comparisons (***p<0.001). (**C**) Concentration-response curves for pharmacochaperoning of SERT-PG$^{601,602}$AA by noribogaine and ECSI#6. Rescued uptake was normalized to uptake velocity measured in parallel in HEK293 cells transiently expressing WT-SERT to account for inter-experimental variations. E$_{MAX}$ and EC$_{50}$ for noribogaine and ECSI#6 were determined by fitting the data to the equation for a rectangular hyperbola (95% CI in parenthesis): for noribogaine and ECSI#6, E$_{MAX}$ was 69.5% (59.2–79.7) and 52.0% (44.6–59.5), and EC$_{50}$ was 0.46 µM (0.15–0.77) and 2.9 µM (1.1–4.7), respectively. The data were obtained in at least three independent experiments carried out in triplicate. The error bars indicate SD. (**D**) Confluent cultures of HEK293 cells transiently expressing SERT-PG$^{601,602}$AA (1 well of a 6-well plate/condition) were treated with either cocaine (30 µM), noribogaine, or ECSI#6 in the indicated concentrations for 24 hr. Untreated cells were taken as negative controls (second lane in the representative blot). Membrane proteins extracted from these cells were denatured and resolved with SDS-PAGE and transferred onto nitrocellulose membranes. The blots were incubated overnight at 4°C with anti-GFP (top) or anti-Gβ (bottom, loading control) antibodies. The immunoreactive bands were detected with fluorescently labeled secondary antibodies. The blot is representative of three independent experiments. (**E**) The intensities of the immunoreactive bands were quantified by densitometry; the ratio of mature (M) to core-glycosylated band (**C**) was corrected for the intensity of the loading control (Gβ). These normalized values (expressed as AU – arbitrary units) were plotted as a function of drug concentration, and fitted to an equation for a rectangular hyperbola. The E$_{MAX}$ and EC$_{50}$ (95% CI in parenthesis) were E$_{MAX}$ = 1.43 AU (1.18–1.68) and 1.42 (1.18–1.65), EC$_{50}$=0.14 µM (0.01–0.35) and 0.44 µM (0.01–1.05) for noribogaines and ECSI#6, respectively. (**F**) Lysates were prepared from HEK293 cells, which expressed wild-type SERT (WT-SERT) and mutant (SERT-PG$^{601-602}$AA) and which were preincubated with the indicated concentrations of cocaine, noribogaine, or ECSI#6 for 24 hr, or from non-transfected HEK-cells (HEK). Aliquots of the lysate (20 µg) were subjected to enzymatic digestion by endoglycosidase H (Endo H). Endo H specifically cleaves core glycans to generate the lower molecular weight deglycosylated (D) from the core-glycosylated species (**C**). Mature glycosylated bands (M) are resistant to the actions of Endo H. The immunoreactive bands were detected as in (**D**). The blot is representative of three independent experiments.

---

quantified with. As expected, pretreatment of cells with a saturating concentration of cocaine did not cause any appreciable increase in the SERT species carrying mature glycans (lane 3, *Figure 5D*). We confirmed the band assignment by enzymatic deglycosylation (*Figure 5F*): the upper bands (labeled M), which appeared in cells incubated in the presence of ECSI#6 and of norbogaine, were resistant to deglycosylation by endoglycosidase H (which cannot cleave mature glycans). In contrast, the core-glycosylated species (labeled C) was susceptible to cleavage by endoglycosidase H resulting in the appearance of the deglycosylated band D.

Human SERT and DAT can substitute for their orthologs in *Drosophila melanogaster* (*Kasture et al., 2018*). Previous studies showed that, when administered to flies via their food, pharmacochaperones can restore the delivery of folding-deficient mutants of DAT to the presynaptic specializations of dopaminergic neurons in vivo (*Kasture et al., 2016*; *Asjad et al., 2017*). Accordingly, we examined the effect of ECSI#6 and the reference compound noribogaine on transporter trafficking by generating transgenic flies. Serotonergic neurons innervate the dorsal fan-shaped body (FB) neuropil in the central brain of *D. melanogaster*. These projections (*Figure 6A–A"*) and the FB6K-type neurons from which they originate in the posterior brain (*Figure 6B–B"*) can be visualized by expressing membrane-anchored GFP (i.e., GFP fused to the C-terminus of murine CD8; *Lee and Luo, 1999*) under the control of TRH-T2A-Gal4. Similarly, when placed under the control of TRH-T2A-Gal4, YFP-tagged wild-type human SERT was expressed in the FB6K-type neurons (*Figure 6C'*) and delivered to the fan-shaped body (*Figure 6C*). In contrast, in flies harboring human SERT-PG$^{601,602}$AA, the transporter was visualized in the soma of FB6K-type neurons (*Figure 6F'*), but the fan-shaped body was devoid of any specific fluorescence (*Figure 6F*). However, if 3-day old male flies expressing human SERT-PG$^{601,602}$AA were fed with food pellets containing 100 µM ECSI#6 or 100 µM noribogaine for 48 hr, fluorescence accumulated to a level, which allowed for delineating the fan-shaped body (*Figure 6G and H*, respectively). This show that ECSI#6 and noribogaine exerted a pharmacochaperoning action in vivo, which partially restored the delivery of the mutant transporter to the presynaptic territory. As expected, in flies harboring wild-type human SERT, feeding of ECSI#6 and noribogaine did not have any appreciable effect on the level of fluorescence in the fan-shaped body (*Figure 6D and E*, respectively).

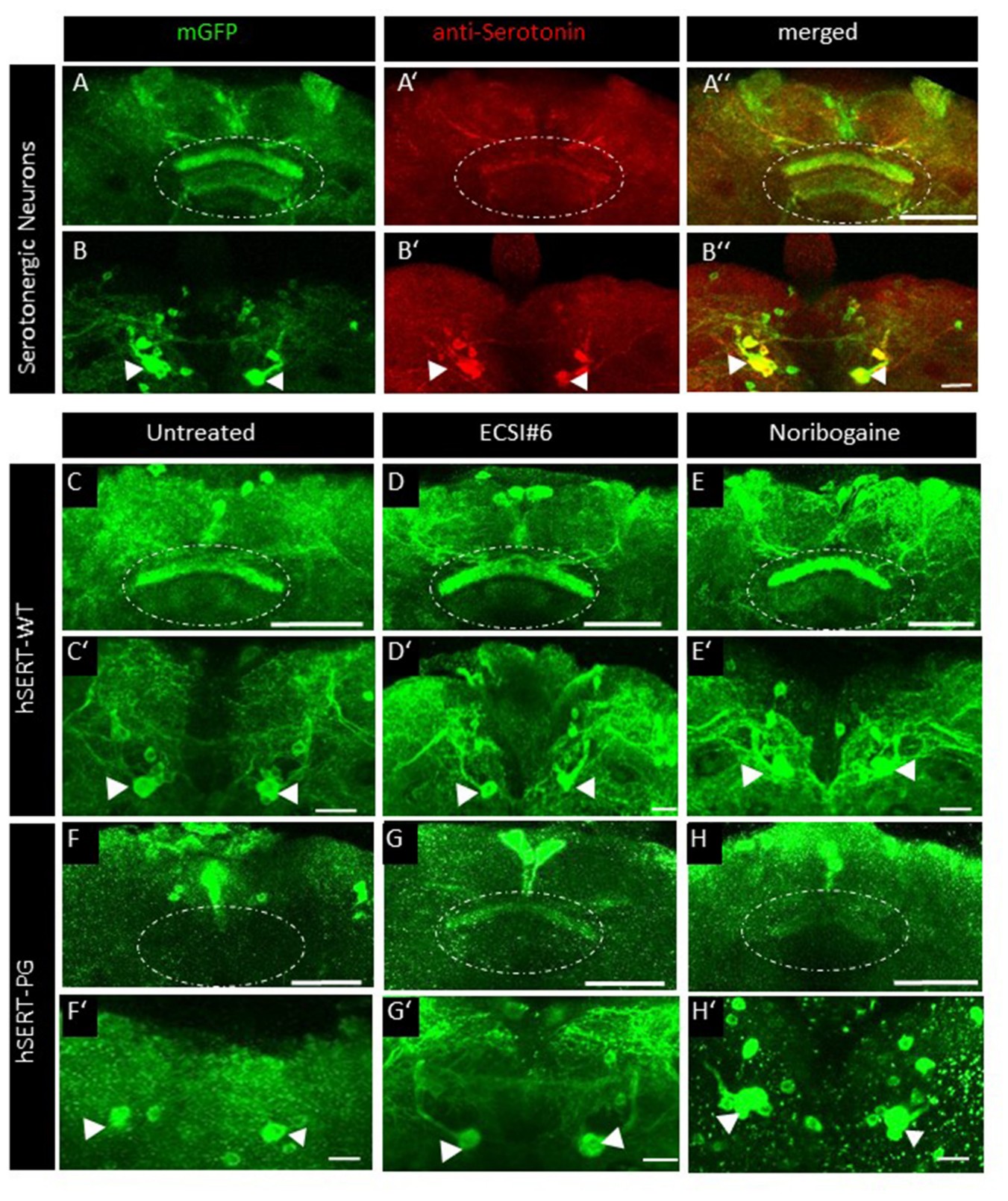

**Figure 6.** ECSI#6 and noribogaine modulate presynaptic expression of hSERT-PG 601,602 AA in the adult fly brain. The top row (**A–A''**) shows the expression of membrane-anchored GFP in the presynaptic compartment of serotonergic neurons in distinct fan-shaped body (FB) layers (dotted circle) co-labeled with an anti-serotonin antibody (red) and the merged image of the presynaptic compartment. (**B–B''**) Cell body clusters of the corresponding serotonergic neurons in the posterior brain (arrowheads). (**C–E'**) Localization of YFP-tagged human wild-type serotonin transporter (hSERT-WT) in the

*Figure 6 continued on next page*

*Figure 6 continued*

FB neuropile (dotted circle in **C–E**) and in cell bodies of the corresponding serotonergic neurons (arrowheads in **C'–E'**) either from 3-day-old male untreated flies (**C, C'**) or flies treated for 48 hr with 100 μM ECSI#6 (**D, D'**) or 100 μM noribogaine (**E, E'**) in their food. (**F–G'**) Localization of YFP-tagged human mutant SERT-PG[601,602]AA (hSERT-PG in the FB neuropile [dotted circle in **F–H**]) and in cell bodies of the corresponding serotonergic neurons (arrowheads in **F'–H'**) either from 3-day-old male untreated flies (**F, F'**) or flies treated for 48 hr with 100 μM ECSI#6 (**G, G'**) or 100 μM noribogaine (**H, H'**) in their food. Images from the brains of these flies were captured by confocal microscopy and compiled with the ImageJ software. Images are single representatives of >10 brains per condition. Scale bars: (**A,C–H**) 50 μm; (**B,C'–H'**) 20 μm. Genotypes: (**A–B**) *UAS-mCD8GFP; TRH^{T2A}-GAL4*. (**C-E'**);*UAS-YFP-hSERT-WT/TRH^{T2A}-GAL4;* (**F-´H'**);;*UAS-YFP-hSERT-PG 601,602 AA/TRH^{T2A}-GAL4*.

## Cocaine, noribogaine, and ECSI#6 bind preferentially to different conformations of SERT

We implemented a simple kinetic model that represents the transport cycle of SERT (reaction scheme in black, *Figure 7A*) to recapitulate the experimental observations on transport inhibition by cocaine, noribogaine, and ECSI#6. In the model, the transport cycle begins with the binding of sodium ($Na^+$) and substrate (S) to SERT to the apo outward-facing state (To). This causes the transporter to isomerize from the substrate-bound outward open (ToNaS) to the substrate-bound inward open (TiNaS) state. Subsequently, $Na^+$ and substrate dissociate on the intracellular side and SERT completes the cycle by anti-porting $K^+$ (TiK →ToK). The reactions outlined in gray in *Figure 7A* represent the possible states of SERT to which an inhibitor can bind.

Inhibition of substrate uptake by cocaine can be modeled as competitive inhibition, where the inhibitor precludes binding of 5-HT to SERT in the ToNa state (*Figure 7B*). In contrast, it is necessary to posit preferential binding of ECSI#6 to the TiK state of SERT for recapitulating the experimental observations by the synthetic data (*Figure 1D* and *Figure 7C*). We extracted from these simulations the $IC_{50}$-values for uptake inhibition by ECSI#6 in the presence of 0.1 and 10 μM 5-HT: the ratio $IC_{50(0.1)}/IC_{50(10)}$ was 20.6, which is reasonably close to the shift of 17.2 observed in *Figure 1D*. This value matches the ratio of occupancy of the TiK state at 10 and 0.1 μM 5-HT (*Figure 7D*). Thus, increasing 5-HT concentrations raise the availability of SERT in the TiK state and this – rather than the affinity of ECSI#6 – accounts for the enhanced ability of ECSI#6 to block 5-HT transport. This interpretation was further substantiated by simulations, where the affinity of ECSI#6 to the TiK state was varied: a variation of the $K_D$ over three orders of magnitude did not alter the $IC_{50(0.1)}/IC_{50(10)}$ ratio (*Figure 7E*). Interestingly the $IC_{50(0.1)}/IC_{50(10)}$ ratios remain high, if ECSI#6 binding is assumed to have a preference to other inward open transporter states (i.e., Ti, TiS, and TiNaS). As expected, if ECSI#6 is assumed to bind preferentially to the outward-open states (i.e., ToK, To, and ToNa), the $IC_{50(0.1)}/IC_{50(10)}$ ratios decrease below 1 (*Figure 7F*). An exception is the high $IC_{50(0.1)}/IC_{50(10)}$ ratio, which is observed, if ECSI#6 binding is assumed to the ToNaS state. This, however, requires binding to an allosteric site, which is unlikely: none of our experimental observations support occupancy of this state by ECSI#6. Thus, the most plausible explanation is the binding of ECSI#6 to the TiK state. We estimated the binding affinity of ECSI#6 for TiK from the synthetic data (*Figure 7G*): a $K_D$ of 12 μM sufficed to reproduce all experimental data. In contrast, it was only possible to emulate the experimental data by assuming that the binding affinities to the other preferred states (i.e., TiS, TiS, TiNaS, and ToNaS, *Figure 7F*) were higher, that is, in the submicromolar range for TiS, TiS, TiNaS or 1 μM for ToNaS (*Figure 7G*). Thus, preferential binding of ECSI#6 to these states (rather than TiK) allows for a possible, but highly unlikely solution.

Finally, we also interrogated the model to account for the shift in the concentration-response curves for uptake inhibition by noribogaine (*Figure 1C*); the shift was less pronounced than that seen for ECSI#6. These observations can be accounted for in the simulations (*Figure 7H*) by assuming preferential binding of noribogaine to both, the ToNa and the TiK states of SERT. If $K_D$ for ToNa is assumed to be 10 times larger than the $K_D$ for TiK, the kinetic model simulations yield an $IC_{50(0.1)}/IC_{50(10)}$ ratio of 5 for noribogaine. This ratio is reasonably close to the ratio of 3, which was observed in the actual experiments. We also systematically varied the ratio of $K_D(ToNa)/K_D(TiK)$ to examine the ratio required to reach a ratio $IC_{50(0.1)}/IC_{50(10)}$ of 17–21, that is, the range seen with ECSI#6. As evident from *Figure 7I*, the $K_D(ToNa)$ must be 100- to 1000-fold larger than $K_D(TiK)$. Hence, based on these calculations, it is appropriate to conclude that ECSI#6 has a much higher preference for binding to TiK than noribogaine. In contrast, the binding mode of noribogaine is mixed. This accounts for the observation

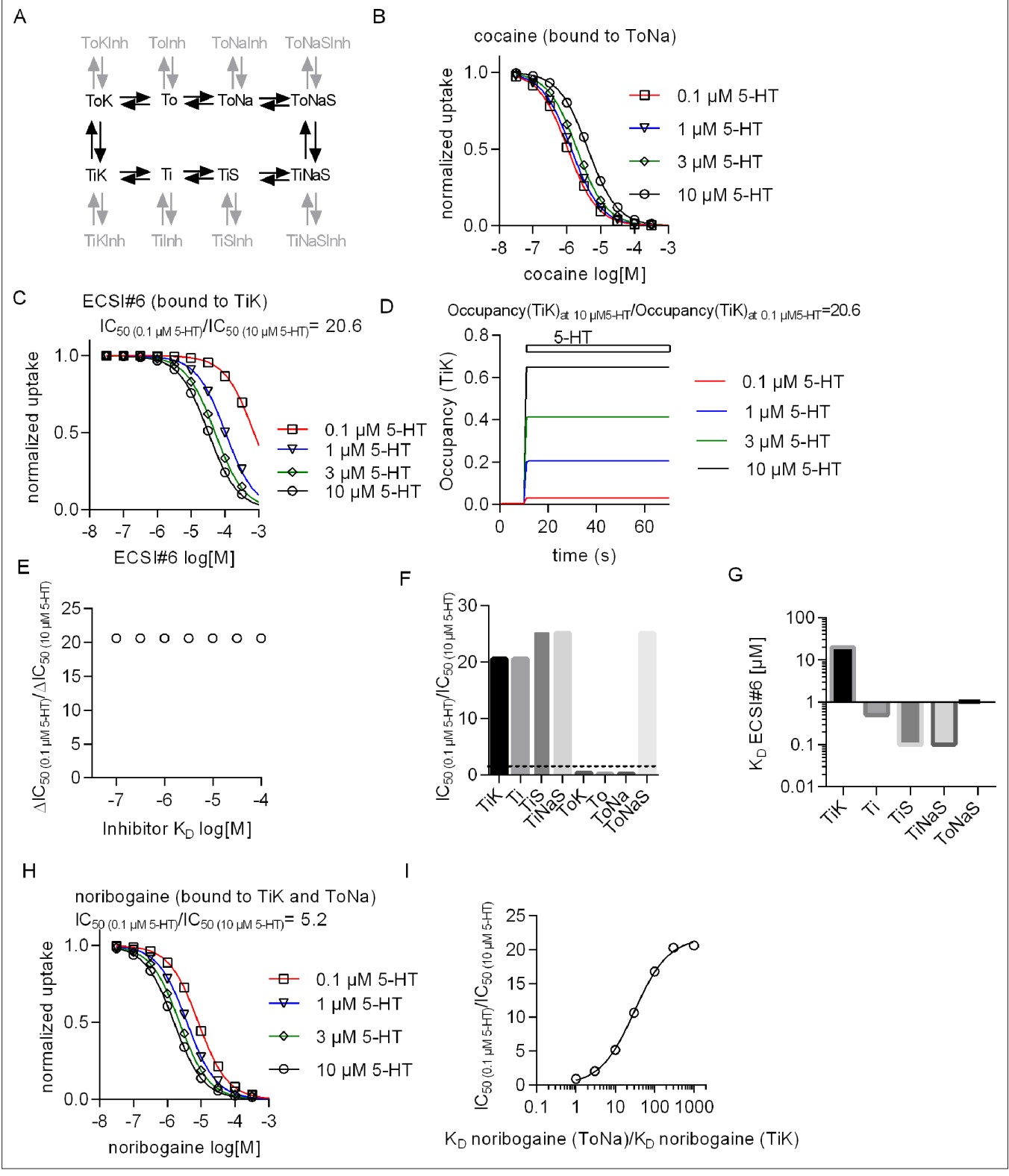

**Figure 7.** Conformational preference of cocaine, noribogaine, and ECSI#6 for binding to serotonin transporter (SERT). (**A**) Reaction scheme (black) of a simple kinetic model of substrate transport by the SERT. The chosen microscopic rate constants were as follows: $k_{on}(K^+)=10^6$ $M^{-1}$ $s^{-1}$, $k_{off}(K^+)=5000$ $s^{-1}$, $k_{on}(Na^+)=10^6$ $M^{-1}$ $s^{-1}$, $k_{off}(Na^+)=1000$ $s^{-1}$, $k_{on}(5-HT)=10^7$ $M^{-1}$ $s^{-1}$, $k_{off}(5-HT)=500$ $s^{-1}$, ToNaS→TiNaS = 60 $s^{-1}$, TiNaS→ToNaS = 75 $s^{-1}$, TiK→ToK = 5 $s^{-1}$, ToK→TiK = 4 $s^{-1}$. The reactions in gray represent possible states where a SERT-specific inhibitor can bind. In the simulation we assumed that all inhibitors bind to SERT with the same association rate (i.e., $k_{on}$(inhibitor)=$10^6$ $M^{-1}$ $s^{-1}$) and we set the $K_D$s to the desired values by adjusting the corresponding dissociation

*Figure 7 continued on next page*

*Figure 7 continued*

rates ($k_{off}$(inhibitor; $K_D = k_{off}/k_{on}$)). A value in the range of $10^6$ $M^{-1}$ $s^{-1}$ is typical for the $k_{on}$ of SERT inhibitors (*Sandtner et al., 2016*; *Hasenhuetl et al., 2015*). (**B**) Simulation of uptake inhibition by cocaine matches experimental data when the preferred state of cocaine binding is to the ToNa state. (**C**) Simulation of uptake inhibition by ECSI#6 matches experimental data when the preferred state of ECSI#6 binding is to the TiK state. The ratio of ECSI#6 $IC_{50}$ in the presence of 0.1 and 10 µM 5-HT ($IC_{50(10)}/IC_{50(10)}$) is equal to 20.6. (**D**) Simulations of occupancy of TiK states with increasing concentrations of 5-HT over time. The ratio of steady-state 5-HT occupancy of TiK at 10 and 0.1 µM is also 20.6. (**E**) The observed shift in the ratio $IC_{50(10)}/IC_{50(10)}$ is not affected, if the affinity of ECSI#6 for SERT is varied. (**F**) Assessment of the $IC_{50(0.1/10)}$ ratio upon assigning binding preference of ECSI#6 to different possible inhibitor-bound states. ECSI#6 shows a clear preference for the inward-facing states of SERT. The notable exception is the ToNaS state, which requires binding to an allosteric site in SERT. (**G**) The true affinity estimates of ECSI#6 were extracted from the simulations by assuming preferential binding to the indicated distinct states of SERT. (**H**) Simulation of uptake inhibition by noribogaine matches the experimental data, if the preferred state of noribogaine binding is to both ToNa and TiK states. (**I**) A ratio of $IC_{50(10)}/IC_{50(10)}$ of 5, which was close to that observed in the actual experiments with noribogaine, can be recapitulated in the simulations, if the $K_D$ of noribogaine to the ToNa state is assumed to be 10-fold higher than the $K_D$ to the TiK state.

that the phenotypic consequence of noribogaine binding is a type of uptake inhibition, which is intermediate between that of cocaine and that of ECSI#6.

We verified this interpretation by simulating the saturation kinetics of 5-HT transport by SERT in the presence of the inhibitor. For noribogaine, the simulations show a reduction in the $V_{max}$ of 5-HT transport with increasing noribogaine concentrations (*Figure 8A*); the concentration-dependent drop in $V_{max}$ was in reasonable agreement with the experimental observations (*Figure 3B*). However, if the $K_M$ of 5-HT transport by SERT was plotted as a function of noribogaine concentration, the analysis revealed an initial drop in $K_M$, which leveled off (*Figure 8C*). This finding is consistent with binding of noribogaine to both, ToNa and TiK state of SERT. The initial drop in $K_M$ is too small to allow for its detection within the experimental inter-assay variation of $K_M$ estimates (*Figure 3E*), while the reduction in $V_{max}$ is readily seen (*Figure 3H*). Hence, based on experimental observations, uptake inhibition by noribogaine is classified as non-competitive. We stress, however, that this neglects the complex binding mode of noribogaine. In contrast, the simulations of substrate transport in the presence of ECSI#6 recapitulated the reduction in the $K_M$ and $V_{max}$ of 5-HT transport by SERT (*Figure 8B and D*), which was seen in the actual experiments summarized in *Figure 3C, F and I*, and thus confirmed the uncompetitive mode SERT of inhibition.

## Discussion

SERT is arguably the best understood SLC transporter (*César-Razquin et al., 2015*). Atomic details are available for several conformations of SERT (*Coleman et al., 2016*; *Coleman and Gouaux, 2018*; *Coleman et al., 2019*). The ligand binding pocket and its coupling to the vestibular binding site have been probed by atomic force microscopy, which defined the energy landscape of the binding interaction at the single molecule level (*Wildling et al., 2012*; *Zhu et al., 2016*). The kinetics of its transport cycle have been extracted from recordings in real time, which also allowed for deducing the cooperativity of substrate and co-substrate binding (*Bhat et al., 2021c*; *Hasenhuetl et al., 2019*; *Bhat et al., 2017*; *Segel, 1975*; *Schicker et al., 2012*; *Sandtner et al., 2014*). The rich pharmacology of SERT comprises typical and atypical inhibitors, partial and full releasers and allosteric modulators (*Hasenhuetl et al., 2019*; *Niello et al., 2020*; *Bhat et al., 2019*). Here, we identify ECSI#6 as a SERT inhibitor with unique properties that are not matched by any of the known ligands: Substrate enhanced the ability of ECSI#6 to inhibit transport resulting in a progressive lowering of the $IC_{50}$ with increasing concentrations of substrate. In a reciprocal manner, ECSI#6 decreased the $K_M$ and the $V_{max}$ of SERT for substrate translocation. This is the hallmark of uncompetitive inhibition. To the best of our knowledge, this is the first report of an uncompetitive inhibitor of SERT and, in fact, of any transporter. Uncompetitive inhibition is rare (*Cornish-Bowden, 1986*). In enzymes, uncompetitive inhibition requires binding of the inhibitor to the enzyme-substrate complex (*Cornish-Bowden, 1986*). Accordingly, the kinetic model also offered allosteric binding of ECSI#6 to the fully loaded outward-facing transporter that is, the ternary complex ToNaS of SERT, substrate and sodium – as one possible solution. However, several arguments rule out this solution: (i) The simulations showed that ECSI#6 must bind with $K_D$ of 1 µM to ToNaS to recapitulate the experimental data. However, in the presence of sub-saturating 5-HT (10 µM), the affinity estimate of ECSI#6 was >30-fold lower (*Figure 2C*). (ii) It is also evident from our observations that ECSI#6 and serotonin were not bound simultaneously to SERT: raising the serotonin

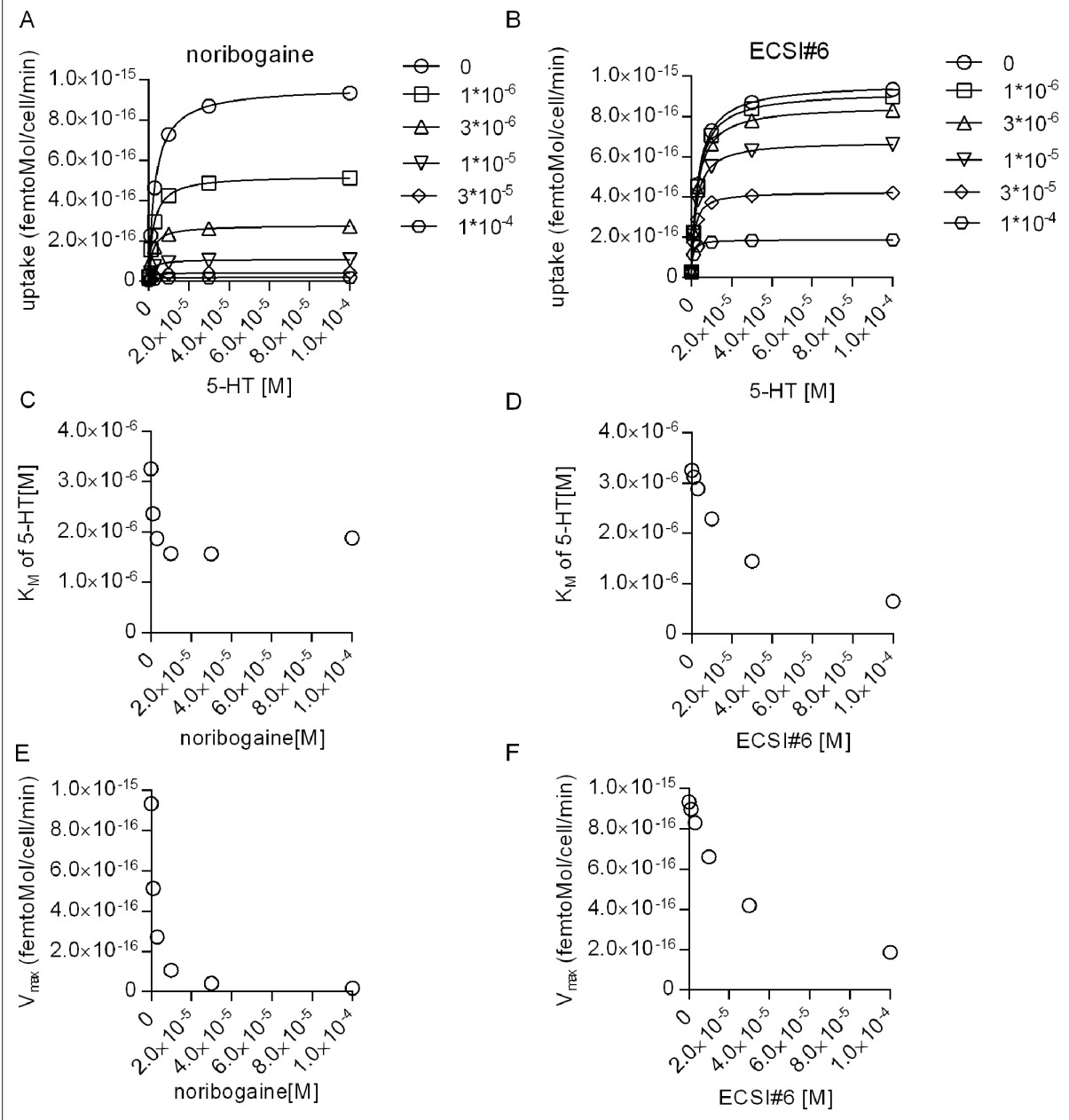

**Figure 8.** Interrogating the kinetic model to analyze the inhibition of substrate transport by serotonin transporter (SERT) by noribogaine and ECSI#6. (**A and B**) Simulations of the saturation kinetics of SERT in the presence of the indicated concentrations of noribogaine (**A**) or ECSI#6 (**B**). The curves were generated using the reaction scheme depicted in *Figure 7A* and the parameters calculated in *Figure 7* for noribogaine and ECSI#6. The $K_M$ of 5-HT uptake by SERT was extracted by fitting the data points to a rectangular hyperbola and plotted as a function of the concentration of noribogaine (**C**) and ECSI#6 (**D**). In (**E**) and (**F**) we show the $V_{max}$ values obtained from the simulation as a function of the noribogaine and ECSI#6 concentration, respectively.

concentration shifted the curves for a displacement of [³H]citalopram by ECSI#6 to the right. Thus, ECSI#6 binds to the same site as serotonin, albeit in a different mode: binding of ECSI#6 was favored in the presence of potassium. The resulting affinity estimate (5 µM, *Figure 2F*) was in the range of the $K_D$-values for the inward-facing potassium-bound conformation of SERT TiK required by the kinetic model to recapitulate the experimental data. (iii) The electrophysiological recordings showed that the block of the steady-state, transport-associated current was imposed by ECSI#6 upon binding to an inward-facing state of SERT. (iv) This conclusion was also confirmed by the other solutions provided by the kinetic model: in these solutions, uncompetitive inhibition by ECSI#6 was accounted for by binding of ECSI#6 to the inward-facing states of SERT. We note that an allosteric action of ECSI#6

– that is, binding to the inward-facing ternary complex TiNaS – was also permitted by the model. However, we consider this unlikely a realistic solution, because – as already pointed out – binding of ECSI#6 and serotonin was mutually exclusive. Hence, we do not consider the analogy of an enzyme-substrate complex, which recruits the inhibitor (*Cornish-Bowden, 1986*), appropriate to understand the mechanisms underlying uncompetitive inhibition of SERT by ECSI#6. In ligand-gated ion channels, uncompetitive inhibition is observed with open channel blockers. This is exemplified by the NMDA-receptor, where dizocilpine maleate/MK-801, memantine, ketamine, and related compounds require the presence of agonists to reach their binding site and to establish blockage of the channel (*Traynelis et al., 2010*). This use-dependent inhibition is more useful as a starting point than the enzyme-substrate complex to understand the mechanisms underlying uncompetitive inhibition of SERT by ECSI#6: in the presence of physiological ion gradients, SERT accumulates in the outward-facing state (*Schicker et al., 2012*). Serotonin is required to drive SERT into the transport cycle, where SERT accumulates in the inward-facing state, because the return step (TiK → ToK, *Figure 7A*) is rate-limiting (*Schicker et al., 2012*). This causes the inhibitory action of ECSI#6 to be use-dependent, that is, enhanced by substrate. Thus, preferential binding of ECSI#6 to the potassium-bound, inward-facing state of SERT (TiK) is sufficient to account for all experimental data.

Our detailed analysis also sheds light on the enigmatic properties of ibogaine. Ibogaine and its more potent analog noribogaine have been classified as non-competitive inhibitors. This mode of inhibition was attributed to their trapping the transporter in the inward-facing conformation (*Jacobs et al., 2007*; *Korkhov et al., 2006*). In fact, the structure of the complex of ibogaine and SERT reveals an inward-facing conformation (*Coleman et al., 2019*). Surprisingly, however, ibogaine binds to SERT from the extracellular side; it cannot gain access to SERT via the inner vestibule (*Burtscher et al., 2018*). Access from the extracellular side also explains, why ibogaine and carbamazepine, which binds to the vestibular (S2) site, can be bound concomitantly (*Sarker et al., 2010*). Finally, sole binding of ibogaine or noribogaine to the inward-facing conformations ought to cause uncompetitive inhibition in a manner similar to ECSI#6. The current observations solve this conundrum: the kinetic model showed that an interaction of noribogaine with both, the sodium-bound, outward-facing state (ToNa) and the potassium-bound, inward-facing state (TiK) of SERT, was required to recapitulate the experimental observations. The kinetic model also revealed the predicted uncompetitive component, which was, however, too small to be detectable in the experimental noise. It is worth pointing out that the overall $K_M$ of substrate transport is a compound parameter, which is affected by both, the rates of individual partial reactions and the occupancy of the individual states in the transport cycle (*Schicker et al., 2021*; *Schicker et al., 2022*). Based on the insights provided by the combination of experimental observations and kinetic modeling, we conclude that, in contrast to ECSI#6, noribogaine has a mixed binding mode.

The endpoint of the folding trajectory of a protein is a minimum energy conformation. The outward- and the inward-facing conformation are equivalent stable folds (*Ponleitner et al., 2022*). The ionic conditions in the ER differ from those at the cell surface. The absence of a sodium gradient favors the inward-facing conformation of SERT and other SLC6 transporters. Thus, it is reasonable to posit that their folding trajectory visits the inward-facing conformation (*Chiba et al., 2014*; *Freissmuth et al., 2017*). This conjecture is supported by the observation that mutations, which trap SERT in the inward-facing state, act as second site suppressors: they restore export from the ER and cell surface delivery of folding-deficient mutants (*Koban et al., 2015*). In addition, many compounds, which act as pharmacochaperones on SERT or DAT, are atypical inhibitors with appreciable affinity for the inward-facing state (*Bhat et al., 2019*; *Bhat et al., 2021b*; *El-Kasaby et al., 2010*; *Bhat et al., 2021a*; *Kasture et al., 2016*; *Asjad et al., 2017*; *Beerepoot et al., 2016*; *Sutton et al., 2022*). Thus, based on its preferential binding to the inward-facing state of SERT, ECSI#6 was predicted to have pharmacochaperoning activity. This prediction was verified with SERT-PG[601,602]AA: pretreatment with ECSI#6 corrected the folding defect of this mutant, because it restored export from the ER and substrate transport by SERT-PG[601,602]AA. In addition, the administration of ECSI#6 to flies resulted in the delivery of the mutants to the presynaptic specialization of the axonal projection to the fan-shaped body. We note, though, that noribogaine was more efficacious as a pharmacochaperone: $E_{max}$ of pharmacochaperoning, that is, the extent to which substrate transport was restored at saturating concentrations – was consistently higher with noribogaine than with ECSI#6. Variable efficacy of pharmacochaperones was noted previously (*Bhat et al., 2021a*; *Sutton et al., 2022*). Efficacy also

depends on the nature of the mutation: individual mutants are stalled at different positions within the energy landscape of the folding trajectory (*Freissmuth et al., 2017*; *El-Kasaby et al., 2010*; *El-Kasaby et al., 2014*; *Koban et al., 2015*). Thus, individual mutants must differ in their susceptibility to pharmacochaperoning. In addition, in a given folding-deficient mutant, closely related compounds differ in $E_{max}$ of pharmacochaperoning (*Bhat et al., 2017*; *Bhat et al., 2021a*; *Sutton et al., 2022*). Thus, pharmacochaperoning efficacy is also an intrinsic property of a compound linked to its chemical structure. Presumably, for a given mutant, variation in pharmacochaperoning efficacy reflects the different ability of individual compounds to stabilize one or several folding intermediates. There are four mechanisms, which account for pharmacochaperoning (*Marinko et al., 2019*): (i) binding to and stabilization of the folded state shifts the folding equilibrium and prevents the backward reaction; (ii) binding to folding intermediates smoothens the energy landscape and precludes stalling of the trajectory in unproductive traps of local minima; (iii) prevention of aggregate formation; and (iv) dissolution of aggregates can maintain or restore the folded state. The latter two mechanisms are immaterial to SERT: misfolded SERT does not form aggregates in the ER (*Anderluh et al., 2014*). The $EC_{50}$ of ECSI#6 for pharmacochaperoning was lower than the concentration required for half-maximum inhibition of substrate uptake at saturating substrate. This is circumstantial evidence for an action of ECSI#6 on folding intermediates. Regardless of the underlying mechanism, because ECSI#6 was more potent a pharmacochaperone than a transport inhibitor, it is an attractive starting point to search for more efficacious compounds with pharmacochaperoning activity. In addition, in vivo, the action of ECSI#6 on serotonergic transmission is likely to differ from typical competitive inhibitors: because of the uncompetitive mode of its inhibition, ECSI#6 is predicted to cause a use-dependent block of SERT. Serotonergic pathways, which when activated, are more susceptible to modulation by ECSI#6. The reverse is true for competitive – that is, surmountable – inhibitors.

## Materials and methods

### Cell culture and materials

The HEK293 cells were purchased from ATCC (# CRL-1573; ATCC, USA) and authenticated by STR profiling at the Medical University of Graz (Cell Culture Core Facility). The cells were cultured in Dulbecco's modified Eagle's medium (DMEM) supplemented with 10% heat-inactivated fetal bovine serum, 0.6 mg $L^{-1}$ penicillin and 1 mg $L^{-1}$ streptomycin and 5 mg $L^{-1}$ plasmocin. These cells were transfected by combining either YFP-tagged WT-SERT or YFP-tagged SERT-PG$^{601,602}$AA with PEI (linear 25 kDa polyethylenimine; Santa Cruz, SC-360988A) at a ratio of 1:3 (w/w) in serum-free DMEM. HEK293 cells stably expressing YFP-tagged SERT, which were used for uptake experiments, were cultured in a medium supplemented with 100 mg $L^{-1}$ geneticin (G418) for clonal selection. HEK293 cells expressing GFP-tagged SERT, which were used for binding experiments and electrophysiology, were cultured in a medium supplemented with 150 mg $L^{-1}$ of zeocin and 6 mg $L^{-1}$ blasticidin. The expression of GFP-SERT was induced by 1 mg $L^{-1}$ tetracycline 24 hr prior to either membrane preparation or electrophysiology. All cells were regularly tested for mycoplasma contamination by 4',6-diamidino-2-phenylindole staining. All antibiotics were purchased from InvivoGen (USA). For the pharmacochaperoning experiments, noribogaine was purchased from Cfm Oskar Tropitzsch GmbH (Marktredwitz, Germany) and ECSI#6 was a kind gift from Dr Sándor Antus (University of Debrecen, Hungary). [³H]5-HT (serotonin, 41.3 Ci mmol$^{-1}$) and [³H]citalopram (80 Ci mmol$^{-1}$) were purchased from PerkinElmer Life Sciences (Rodgau, Germany). The scintillation mixture (Rotiszint eco plus) was purchased from Carl Roth GmbH (Karlsruhe, Germany). Cell culture media were obtained from Sigma-Aldrich (St Louis, MO, USA). The anti-GFP antibody (rabbit, ab290) was from Abcam (Cambridge, UK). An antibody raised against an N-terminal peptide of the G protein β subunit (*Hohenegger et al., 1996*) was used to verify comparable loading of lanes. The secondary antibody (donkey anti-rabbit, IRDye 680RD) was obtained from LI-COR Biotechnology GmbH (Bad Homburg, Germany). All other chemicals were of analytical grade.

### [³H]Citalopram binding

Membranes were prepared from HEK293 cells stably expressing GFP-SERT. The cells were washed twice in phosphate-buffered saline (PBS) (137 mM NaCl, 2.7 mM KCl, 4.3 mM $Na_2HPO_4$, 1.5 mM $KH_2PO_4$, pH adjusted to 7.4), harvested and centrifuged at 2000 rpm for 10 min. The pellets were resuspended in a buffer containing 20 mM HEPES, 2 mM $MgCl_2$, 1 mm EDTA, pH adjusted to 7.4

with NaOH, subjected to freeze-thaw cycles in liquid nitrogen and sonicated on ice three times for 10 s with 30 s intervals. Following another centrifugation step, the whole-cell membrane pellets were resuspended in buffer and protein concentration was estimated by dye binding (Coomassie Brilliant Blue R-250; Bio-Rad, USA). The binding reaction was carried out in a final volume of 0.1 mL of $Na^+$-containing buffer (20 mM Tris-HCl, pH 7.4, 1 mM EDTA, 2 mM $MgCl_2$, 120 mM NaCl) containing either 0, 1, or 10 µM cold 5-HT, membranes (2.5 µg assay$^{-1}$), [$^3$H]citalopram (3 nM), and the logarithmically spaced concentrations of cocaine (0.1–100 µM), noribogaine (0.1–100 µM), and ECSI#6 (0.1–100 µM) at 20°C for 1 hr. Binding reactions were also done in the presence of 120 mM $K^+$ buffer (buffer composition: 20 mM Tris-HCl, pH 7.4, 1 mM EDTA, 2 mM $MgCl_2$, 120 mM KCl; 4 mM $Na^+$ were present from the carry-over of HEPES·NaOH and from Na·EDTA). For binding reactions under these conditions, 7–8 µg of membranes were used to improve the dynamic range of binding. The binding reactions were terminated by harvesting the membranes on glass fiber filters precoated with polyethylenimine and rapid washing with ice-cold wash buffer (10 mM Tris·HCl, pH 7.4, 120 mM NaCl, 2 mM $MgCl_2$). The radioactivity trapped on the filters was quantified by liquid scintillation counting. Nonspecific binding was measured in the presence of 10 µM paroxetine.

## [$^3$H]5-HT uptake

For all uptake assays, HEK293 cells expressing wild-type human YFP-tagged SERT were seeded on poly-D-lysine-coated 96-well plates at a density of 30,000 cells well$^{-1}$. After 24 hr, the medium was removed and washed once with Krebs-HEPES buffer (10 mM HEPES·NaOH, pH 7.4, 120 mM NaCl, 3 mM KCl, 2 mM $CaCl_2$, 2 mM $MgCl_2$, and 2 mM glucose). For uptake inhibition assays, logarithmically spaced concentrations of cocaine (0.1–100 µM), noribogaine (0.1–100 µM), or ECSI#6 (0.3–300 µM) were prepared in a buffer containing either 0, 1, or 10 µM cold 5-HT and added to the washed cells for 10 min as a preincubation step (50 µL assay$^{-1}$). The same logarithmically spaced concentrations of the three compounds were prepared in a buffer containing either 0.2 µM [$^3$H]5-HT or 0.2 µM [$^3$H]5-HT with 1 µM cold 5-HT or 0.2 µM [$^3$H]5-HT with 10 µM cold 5-HT. After preincubation for 10 min, the radioactivity was added for 1 min to achieve a final volume of 100 µl well$^{-1}$ and a final [$^3$H]5-HT concentration of 0.1 µM. The reaction was terminated after 1 min by aspiration of the reaction medium followed by a single wash with ice-cold buffer. For the determination of $K_M$ and maximal velocity $V_{max}$ of 5-HT transport by SERT, YFP-SERT cells were preincubated for 10 min with 50 µL of either buffer in the absence and presence of either cocaine (3, 10, or 30 µM), noribogaine (1, 3, or 10 µM) or ECSI#6 (10, 30, or 100 µM). After the preincubation, 5-HT saturation experiments were undertaken by adjusting the specific activity of 0.1 µM [$^3$H]5-HT with unlabeled 5-HT to vary between 50 cpm fmol$^{-1}$ and 50 cpm pmol$^{-1}$ as follows: logarithmically spaced concentrations of cold 5-HT were prepared in a buffer containing either twice the indicated concentrations of [$^3$H]5-HT in the absence and presence of cocaine (3, 10, or 30 µM), noribogaine (1, 3, or 10 µM), or ECSI#6 (10, 30, or 100 µM). These solutions (50 µL) were added for 1 min to achieve a final volume of 100 µl well$^{-1}$ and a final [$^3$H]5-HT concentration of 0.1 µM. The reaction was terminated after 1 min by aspiration of the reaction medium followed by a single wash with ice-cold buffer.

For uptake assays determining the functional rescue of mutant transporter, HEK293 cells were transfected with either YFP-tagged SERT-PG[601,602]AA or YFP-tagged WT-SERT plasmids. Transfected cells were seeded on poly-D-lysine-coated 96-well plates at a density of ~60–80,000 cells well$^{-1}$ in either the absence or presence of increasing concentrations (0.3–100 µM) of either cocaine, noribogaine, or ECSI#6. After 24 hr, the cells were washed four times with Krebs-MES buffer (10 mM 2-(N-morpholino)ethanesulfonic acid, pH 5.5, 120 mM NaCl, 3 mM KCl, 2 mM $CaCl_2$, 2 mM $MgCl_2$, and 2 mM glucose) in a 10 min interval and once with Krebs-HEPES (pH 7.4) buffer. The cells were subsequently incubated with 0.2 µM of [$^3$H]5-HT for 1 min and the uptake assay was carried out as outlined above. Nonspecific uptake for all uptake experiments was defined in the presence of 10 µM paroxetine. After the uptake reaction, the cells were then lysed with 1% SDS to release the retained radioactivity, which was quantified by liquid scintillation counting.

## Immunoblotting

HEK293 cells were transiently transfected with plasmids encoding either WT-YFP-SERT or YFP-SERT-PG[601,602]AA. Approximately 1.5–2×10$^6$ of these transfected cells were seeded in six-well plates in the presence of either cocaine, noribogaine, or ECSI#6 (0.3–30 µM). After 24 hr, cells were washed thrice

with ice-cold PBS, detached by mechanical scraping, and harvested by centrifugation at 1000 × *g* for 5 min. The cell pellet was lysed in a buffer containing Tris·HCl, pH 8.0, 150 mm NaCl, 1% dodecyl maltoside, 1 mm EDTA, and protease inhibitors (Complete, Roche Applied Science). This soluble protein lysate was separated from the detergent-insoluble material by centrifugation (16,000 × *g* for 15 min at 4°C). An aliquot of this lysate (20 µg) was mixed with a sample buffer containing 1% SDS and 20 mM DTT, denatured at 45°C for 30 min and resolved in denaturing polyacrylamide gels. After protein transfer onto nitrocellulose membranes, the blots were probed with an antibody against GFP (rabbit, ab290) at a 1:3000 dilution overnight. This immunoreactivity was detected by fluorescence detection using a donkey anti-rabbit secondary antibody at 1:5000 dilution (IRDye 680RD, LICOR). The lower part of the blot was also probed with the antibody recognizing the G protein β subunits to verify equal loading. For enzymatic deglycosylation, detergent lysates (20 µg) were denatured at 100°C for 10 min, followed by incubation at 37°C in the presence and absence of endoglycosidase H (New England Biolabs) for 2 hr. Samples were resolved by electrophoresis on denaturing polyacryl-amide gels (7% monomer concentration); immunoblotting was done as described above.

## Whole-cell patch clamp recordings

HEK293 cells stably expressing wild-type GFP-tagged human SERT were seeded at low density on poly-D-lysine-coated dishes. Twenty hr after seeding, these cells were subjected to patch clamp recordings in the whole-cell configuration. In most instances cells were continuously maintained in an external solution containing 140 mM NaCl, 3 mM KCl, 2.5 mM CaCl$_2$, 2 mM MgCl$_2$, 20 mM glucose, and 10 mM HEPES (pH adjusted to 7.4 with NaOH) and the drugs were diluted therein. The internal solution in the patch pipette contained 133 mM potassium gluconate (CH$_2$OH(CHOH)$_4$COOK), 5.9 mM NaCl, 1 mM CaCl$_2$, 0.7 mM MgCl$_2$, 10 mM HEPES, 10 mM EGTA (pH adjusted to 7.2 with KOH). Drugs were applied using a 4-tube ALA perfusion manifold (NPI Electronic GmbH, Germany) and a DAD-12 superfusion system (Adams & List, Westbury, NY, USA) allowing for complete solution exchange around the cells within 100 ms. Current amplitudes and associated kinetics were quantified using Clampfit 10.2 software. Passive holding currents were subtracted, and the traces were filtered using a 100 Hz digital Gaussian low-pass filter.

## *Drosophila* genetics and drug treatment

The transgenic UAS reporter lines for YFP-tagged human wild-type SERT and SERT-PG[601,602]AA were generated using the pUAST-attB vector (gift from Drs Bischof and Basler, University of Zürich) and injected into embryos of ZH-86Fb flies (Bloomington stock no. 24749). Positive transformants were isolated and selected. TRH-T2A-Gal4 (Bloomington stock no. 84694) was used to drive the expression of transporters in serotonergic neurons. Three-day-old male; TRH-T2A-Gal4/UAS-YFP-hSERT-WT; or TRH-T2A-Gal4/UAS-YFP-hSERT- PG[601,602]AA; flies were treated with food supplemented with 100 µM noribogaine or 100 µM ECSI#6 for 48 hr. The brains of these flies were then imaged using confocal microscopy. All flies were kept at 25°C, and all crosses were performed at 25°C.

## Immunohistochemistry and imaging

Adult fly brains were dissected in PBS and fixed in 2% paraformaldehyde in PBS for 1 hr at room temperature. Brains were then washed three times in 0.1% Triton X-100 in PBS for 20 min on a shaker. Blocking was performed in 10% goat serum for 1 hr at room temperature on a shaker. Brains were then incubated in primary antibody overnight in PBS containing 3% BSA and 0.3% Triton X-100 at 4°C on a shaker. The rabbit polyclonal IgG directed against GFP (1:1000 dilution; A-11122, Invitrogen) and anti-serotonin antibody (S5545, Sigma-Aldrich) were used as primary antibodies. After three washes for 20 min with PBS containing 0.1% Triton X-100, the brains were incubated overnight at 4°C with a secondary antibody in PBS containing 0.3% Triton X-100 on a shaker. Alexa Fluor 488- or 568-labeled goat anti-rabbit IgG (1: 500 or 1:300 Invitrogen) was used as a secondary antibody. Following incubation with a secondary antibody, the brains were washed three times with PBS containing 0.1% Triton X-100 and were mounted using Vectashield (Vector Laboratory, Burlingame, CA, USA). Images were captured on a Leica SP5II confocal microscope with 20-fold magnification. Z-stack images were scanned at 1.5 µm section intervals with a resolution of 512×512 pixels. Images were processed with ImageJ.

## Kinetic modeling

A kinetic model of the transport cycle of SERT was built based on the reaction scheme in *Figure 7A*. This scheme is simplified because (i) it ignores $Cl^-$ and proton binding to SERT, (ii) it does not account for voltage dependence, and (iii) it assumes a sequential instead of a random binding order of $Na^+$ and substrate. These simplifications were justified by the fact that in the experiments, which we simulated we did not change the $Cl^-$/proton concentrations in the bath or the electrode solution, we did not apply voltage jumps and we used saturating concentrations of $Na^+$ throughout. The time-dependent changes in state occupancies were evaluated by numerical integration of the resulting system of differential equations using the Systems Biology Toolbox (*Schmidt and Jirstrand, 2006*) in Matlab 2019b (The MathWorks, Inc, Natick, MA, USA). We used this model to evaluate how state-dependent binding of an inhibitor impinges on substrate uptake. To this end, we allowed for the inhibitor to bind to each individual state specified in the reaction scheme at a time. The rate of substrate uptake was modeled as: $TiS*k_{off}-Ti*k_{on}*S_{in}$, where $TiS$ and $Ti$ are the occupancies of the substrate-bound and apo inward-facing conformation, respectively, and $k_{off}$ and $k_{on}$ the dissociation and association rate of substrate and $S_{in}$ the intracellular substrate concentration. The numeric values used to parameterize the microscopic rate constants of the model can be found in the legend of *Figure 7A*.

## Acknowledgements

We thank Sándor Antus (University of Debrecen, Hungary) for the generous gift of ECSI#6. This work was supported by grants from the Vienna Science and Technology Fund/WWTF (LSC17-026 to MF) and from the Austrian Science Fund/FWF (P31255-B27 and P31813 to SS and WS, respectively).

## Additional information

### Funding

| Funder | Grant reference number | Author |
| --- | --- | --- |
| Austrian Science Fund | P31813 | Walter Sandtner |
| Austrian Science Fund | P31255-B27 | Sonja Sucic |
| Vienna Science and Technology Fund | LSC17-026 | Michael Freissmuth |

The funders had no role in study design, data collection and interpretation, or the decision to submit the work for publication.

### Author contributions

Shreyas Bhat, Conceptualization, Data curation, Formal analysis, Investigation, Methodology, Writing - original draft, Writing - review and editing; Ali El-Kasaby, Danila Boytsov, Data curation, Formal analysis, Investigation, Methodology; Ameya Kasture, Data curation, Formal analysis, Investigation, Methodology, Writing - original draft, Writing - review and editing; Julian B Reichelt, Data curation; Thomas Hummel, Resources, Supervision, Funding acquisition, Methodology; Sonja Sucic, Resources, Funding acquisition, Methodology; Christian Pifl, Conceptualization, Resources, Investigation, Methodology, Writing - review and editing; Michael Freissmuth, Conceptualization, Resources, Supervision, Funding acquisition, Methodology, Writing - original draft, Writing - review and editing; Walter Sandtner, Conceptualization, Software, Formal analysis, Supervision, Funding acquisition, Methodology, Writing - original draft, Project administration, Writing - review and editing

### Author ORCIDs

Shreyas Bhat http://orcid.org/0000-0001-7019-9180
Thomas Hummel http://orcid.org/0000-0001-8108-9307
Walter Sandtner http://orcid.org/0000-0003-3637-260X

### Decision letter and Author response

Decision letter https://doi.org/10.7554/eLife.82641.sa1
Author response https://doi.org/10.7554/eLife.82641.sa2

# Additional files

## Supplementary files
• MDAR checklist

## Data availability
We uploaded original data onto Dyrad. The generated DOI is: https://doi.org/10.5061/dryad.fqz612jvz.

The following dataset was generated:

| Author(s) | Year | Dataset title | Dataset URL | Database and Identifier |
|---|---|---|---|---|
| Sandtner W | 2022 | A mechanism of uncompetitive inhibition of the serotonin transporter | https://doi.org/10.5061/dryad.fqz612jvz | Dryad Digital Repository, 10.5061/dryad.fqz612jvz |

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
