## [Editor Report]

This study presents the important finding of an unusual uncompetitive inhibitor (ECSI#6) of the serotonin transporter SERT that removes the neurotransmitter serotonin from the synaptic cleft. Through careful and comprehensive analysis, the authors convincingly show that the molecule most likely binds to the inward-facing and K^+^-bound state and that it assists in folding and targeting the transporter. The work will be of interest to those engaged in biophysical analyses of the serotonin transporter, and colleagues developing pharmacological chaperoning strategies for transporters in general.

---

## [Decision Letter]

**Decision letter after peer review:**

Thank you for submitting your article "A mechanism of uncompetitive inhibition of the serotonin transporter" for consideration by *eLife*. Your article has been reviewed by 3 peer reviewers, and the evaluation has been overseen by a Reviewing Editor and Kenton Swartz as the Senior Editor. The following individual involved in the review of your submission has agreed to reveal their identity: Christof Grewer (Reviewer #3).

The reviewers have discussed their reviews with one another, and the Reviewing Editor has drafted this to help you prepare a revised submission. In general, the reviewers found the study to be interesting and compelling. However, the following clarifications, and in some cases additional data, would make the study more complete.

Essential revisions:

1) The authors suggest that serotonin and ECSI#6 cannot bind simultaneously to the transporter, however, no direct evidence for this conclusion is provided. Either provide data to support this claim or revise the text for a more balanced interpretation.

2) The question remains as to how ECSI#6 accesses the inward-facing binding site. Is this compound expected to permeate across the membrane, or is it slowly transported across and then binds to the intracellular side? To elaborate on this, studies including ECSI#6 in the recording electrode may provide further information on this point.

3) Please elaborate on why displacement experiments were not carried out with cocaine. Since cocaine is a competitive inhibitor but does not induce transport (i.e. doesn't induce the formation of the inward-facing conformation), it should act in a competitive mechanism with ECSI#6.

4) Dose-response relationships for the electrophysiological experiments would be a useful alternative method to validate the radiotracer flux data. Either provide some experimental data on this or explain why these experiments are not possible.

5) In the in-vivo *Drosophila* imaging data, the SERT fluorescence with WT protein is strong, but there is no clear fluorescence in the drug-treated image from the PG mutant. In this context, it is expected that there would be additional intracellular staining for ER-resident SERT. If the cell bodies of these cells are elsewhere this should be clearly pointed out. Please explain or provide additional images to clarify.

6) The Western blot demonstrating enhanced glycosylation in the presence of Noribogaine or ECSI#6 could be strengthened. The increased band at a high molecular weight that the authors attribute to the fully glycosylated form is visible. However, this smear, and the band below, look quite different from those in the blot shown in the El-Kasaby et al. reference, raising concerns that the band could be aggregated or dimerized protein rather than a glycosylated form. This concern could easily be addressed by control experiments with appropriate glycosidases, as shown in the reference.

6) Please address the author recommendations listed by the reviewers below.

*Reviewer #1 (Recommendations for the authors):*

Figure 1B. The fit for 20 μm 5-HT should be dotted lines. Dashes are shown.

Line 296: the explanation of why uptake leads to accumulation of the K^+^-bound intermediate could be improved for the benefit of a non-expert reader, especially since this relates to a central conclusion of the work.

Figure 3G, H, I y-axis should be labeled as Vmax(units), just as those above are labeled "Km(unit)".

Line 561. It should be explicitly mentioned that To is the outward-facing apo state.

Line 586-590. This point is confusingly written and I didn't understand. It should be clarified.

*Reviewer #2 (Recommendations for the authors):*

- I would like to have access to the raw data of the experiments.

- The authors use a lot of text to explain their findings and interpretation, which I like. Nonetheless, in places, I find the paper hard to read, which may be caused by the choice of wording. For instance: line 296, "accumulation", Line 309-130, "accordingly".…appreciable binding can be measured, line 395: it is unclear how continuous cycling would lead to a current for an electroneutral transporter. It would help to explicitly mention that the current is uncoupled.

*Reviewer #3 (Recommendations for the authors):*

Figure 1A: Should the pKa of these inhibitors be given?

Whenever the term "affinity" is used (except in binding studies), it probably should be replaced by "apparent affinity".

Figure 2: It would be useful if panels A-C and D-F would have the same x-axis scale, so the reader can more easily compare the IC50 values in the absence and presence of K^+^.

Line 302: For the reader unfamiliar with SERT pharmacology, it should be explained why citalopram was used for the binding inhibition experiments.

Figure 3 D-F: The plots should be made in a regular x-y fashion, with a linear x (concentration) axis scale. That would make it easier to see if the cocaine data adhere to the regular competitive enzyme inhibition mechanism equation, in which Km is expected to increase in a linear fashion with inhibitor concentration. My feeling is that the slope of that curve is too shallow for cocaine to account for the full competitive mechanism.

Line 393: It is evident from the data that cocaine and noribogaine induce a peak current, but ECSI#6 does not. However, the authors do not explain why this is the case. This seems to be an important detail that could further inform on the inhibition mechanism.

Figure 4 D-E: Linear kapp vs concentration relationships are typically observed in situations where ligand binding is rate limiting, often under diffusion control. From the slope of the relationship, the bimolecular rate constant for binding can be obtained. What is this value? I assume that it is far below a diffusion-controlled rate constant?

Figure 5D: This Western blot is not very convincing. (1) The band at 110 kD is very weak, and this reviewer has trouble reconciling the bands with the data shown in Figure 5E. (2) What is the band at 60 kD? This blot needs further explanation.

Figure 7 legend: No information is given on the microscopic rate constants for inhibitor binding/dissociation. Are these assumed to be in equilibrium with respect to the other transport steps?

Figure 8: The plots of simulated vmax vs. inhibitor concentration should also be shown.

---

## [Author Response]

Essential revisions:1) The authors suggest that serotonin and ECSI#6 cannot bind simultaneously to the transporter, however, no direct evidence for this conclusion is provided. Either provide data to support this claim or revise the text for a more balanced interpretation.

We assessed this point by plotting the data in Figure 2A,B,C as Dixon plots in (the new) panels D,E,F of Figure 2. We refer the reader to Segel’s textbook on enzyme kinetics (new ref. 18) on using Dixon plots in the presence of two inhibitors. The pertinent description is on p. 9, lines 12-22 and reads as follows: “We transformed the data summarized in Figures 2A-C by plotting the reciprocal of bound radioligand as a function of inhibitor concentration to yield Dixon plots (Figure 2D-F): the x-intercept corresponds to -IC50 of the inhibitor [18]. Thus, Dixon plots allow for differentiating mutually exclusive from mutually non-exclusive binding, if one inhibitor (i.e., cocaine, noribogaine or ECSI#6) is examined at a fixed concentration of the second inhibitor (i.e., serotonin) [18]: if binding of the two inhibitors is mutually non-exclusive, a family of lines of progressively increasing slope, which intersect at -IC50, is to be seen. In contrast, if the two inhibitors bind to the same site, the slope of the inhibition curves is not affected and the xintercept (i.e, -IC50 of the variable inhibitor) is shifted to more negative values. It is evident from Figure 2D-E that the presence of 1 and 10 µM serotonin progressively shifted the (expected) x-intercept for cocaine (Figure 2D), noribogaine (Figure 2E) and ECSI#6 (Figure 2D). Thus, binding to SERT of serotonin and of these three ligands was mutually exclusive.” Based on the Dixon plots, we feel that our conclusion is justified, i.e., binding of serotonin and ECSI#6 (and of the other ligands) is mutually exclusive.

2) The question remains as to how ECSI#6 accesses the inward-facing binding site. Is this compound expected to permeate across the membrane, or is it slowly transported across and then binds to the intracellular side? To elaborate on this, studies including ECSI#6 in the recording electrode may provide further information on this point.

We did the suggested experiments: the data are summarized in panel I of Figure 4 and described in the first paragraph on p. 15, which also explains why this experiments is possibly inconclusive due to the high diffusivity of ECSI#6:

“Figure 4I shows representative traces of 5-HT induced currents recorded from SERT expressing cells in the absence (in blue) and presence of 100 µM ECSI#6 (in red) in the electrode solution: when thus applied from the intracellular side, ECSI#6 did not cause an appreciable current block. The right-hand panel summarizes the current amplitude obtained from cells measured in the absence (blue open circles) and presence of intracellular ECSI#6 (open circles in red). These data seem to indicate that ECSI#6 binds to SERT from the extracellular side. Yet this conclusion can be challenged based on the following consideration: in earlier experiments, ibogaine, the parent compound of noribogaine, was found to block HERG channels when applied from the bath solution but failed to do so when added to the electrode solution [27]. However, at a lower intracellular pH (i.e., pH 5.5), ibogaine gained the ability to inhibit HERG from the intracellular side (i.e., via application through the electrode). Conversely, ibogaine was less effective when applied to an acidified bath solution. These observations led to the conclusion that ibogaine blocked HERG from the cytosolic side: because the molecule in its neutral form was so diffusive, a low intracellular pH was required to force its protonation and thus preclude diffusion from the interior of the cell. ECSI#6 is presumed to also be very diffusible given its estimated logP value and polar surface area of 2.48 and 66 Å2, respectively. However, ECSI#6 harbors an amide nitrogen (see Figure 1A) and thus remains neutral in the experimentally accessible pH range. Hence, it is not possible to verify to which side of SERT it binds.”

3) Please elaborate on why displacement experiments were not carried out with cocaine. Since cocaine is a competitive inhibitor but does not induce transport (i.e. doesn't induce the formation of the inward-facing conformation), it should act in a competitive mechanism with ECSI#6.

We did not quite understand this comment from reviewer#1, because displacement experiments were done with cocaine (Figure 2A, new Figure 2G/previous Figure 2D). However, if the reviewer questions why we do not use cocaine rather than 5-HT, in the three-way competition experiment, it is precisely, because we wanted to compare the action/binding mode of ECSI#6 to that of cocaine.

4) Dose-response relationships for the electrophysiological experiments would be a useful alternative method to validate the radiotracer flux data. Either provide some experimental data on this or explain why these experiments are not possible.

These experiments were done and are shown in (the new) panels G and H of Figure 4; the pertinent description is in the second paragraph of p. 14 and reads:

“The protocol depicted in Figure 4B can also be used to gauge the apparent affinity of ECSI#6 for SERT in the presence of 5-HT. Plotted in Figure 4G is the block of the serotonin-induced current as a function of the co-applied ECSI#6 concentration. The current was evoked by a saturating concentration of 5-HT (30µM) and inhibited by 3, 10, 30 and 100 µM co-applied ECSI#6, respectively (the inset in Figure 4G shows representative current traces). A fit of an inhibition curve to the data points yielded an IC50 value of approx. 5 µM. This value was lower but still in reasonable agreement, with the IC50 obtained in the radioligand uptake assay for the condition where the 5-HT concentration had been saturating (cf. dashed line in Figure 1C; 10 µM 5-HT). In the uptake assay the IC50 value of ECSI#6 dropped to about 0.5 mM, in the presence of a low 5-HT concentration (i.e., 0.1 µM). In contrast to uptake experiments, electrophysiological recordings also allow for assessing the apparent affinity of ECSI#6 for SERT in the absence of the substrate. This can be achieved by employing the protocol depicted in Figure 4H (see representative current traces on the left-hand side): we first applied 30 µM 5HT to a cell expressing SERT for 0.5 s to elicit a peak current (i.e., a control pulse). We then reapplied 30 µM 5-HT after a superfusing the cell with 100 µM ECSI#6 for 1 s (second upper trace in panel H). We chose this time period because it had been sufficient to allow for full current block in the other protocol (see Figure 4G): the amplitude of the peak current following pre-application of 100 µM ECSI#6 was essentially identical to the prior control pulse. When we pre-applied 100 µM ECSI#6 for a longer period (i.e., 3 s) the amplitude of the two peak currents also remained the same (cf. lower traces in panel H). The right-hand panel shows the summary of several experiments. Plotted in the graph is the ratio of the second and first pulse, which was always close to one. We previously used this protocol to assess the binding kinetics of cocaine, methylphenidate and desipramine on SERT and DAT. Pre-application of these inhibitors consistently led to a concentration dependent reduction in the peak current amplitude of the second pulse in comparison to the first [23]. The lack of inhibition, thus, indicates that the affinity of ECSI#6 in the absence of 5-HT is low. To obtain estimates for the affinity of ECSI# for SERT in the absence of 5-HT we would need to apply this compound at much higher concentrations. This, however, is not possible, because ECSI#6 is poorly soluble in aqueous solutions (i.e., max. 0.03 mg/ml).”

5) In the in-vivo *Drosophila* imaging data, the SERT fluorescence with WT protein is strong, but there is no clear fluorescence in the drug-treated image from the PG mutant. In this context, it is expected that there would be additional intracellular staining for ER-resident SERT. If the cell bodies of these cells are elsewhere this should be clearly pointed out. Please explain or provide additional images to clarify.

We have modified Figure 6 to include, in all instances, images of the posterior brain, where the neurons (FB6K) reside, from which the serotonergic projections originate. These images visualize expression of membrane-anchored GFP (mCD8GFP; in panel B), immunoreactivity of serotonin (panel B’), wild type SERT (panels C’,D’,E’) and mutant SERT-PG601,602AA (panels F’,G’,H’) in the soma. The description of these panels has been added to the pertinent sentences starting on p. 20, line 6 from bottom to the end of end of the first paragraph p. 21, which read:

“These projections (Figure 6A-A’’) and the FB6K-type neurons, from which they originate in the posterior brain (Figure 6B-B’’) can be visualized by expressing membrane-anchored GFP (i.e. GFP fused to the C-terminus of murine CD8; [36]) under the control of TRH-T2A-Gal4. Similarly, when placed under the control of TRH-T2A-Gal4, YFP-tagged wild-type human SERT was expressed in the FB6K-type neurons (Figure 6C’) and delivered to the fan-shaped body (Figure 6C). In contrast, in flies harboring human SERT-PG601,602AA, the transporter was visualized in the soma of FB6K-type neurons (Figure 6F’), but the fan-shaped body was devoid of any specific fluorescence (Figure 6F). However, if three-day old male flies expressing human SERTPG601,602AA were fed with food pellets containing 100 µM ECSI#6 or 100 µM noribogaine for 48 h, fluorescence accumulated to a level, which allowed for delineating the fan-shaped body (Figure 6G and H, respectively). This show that ECSI#6 and noribogaine exerted a pharmacochaperoning action in vivo, which partially restored the delivery of the mutant transporter to the presynaptic territory. As expected, in flies harboring wild-type human SERT, feeding of ECSI#6 and noribogaine did not have any appreciable effect on the level of fluorescence in the fan-shaped body (Figure 6D and E, respectively). “

6) The Western blot demonstrating enhanced glycosylation in the presence of Noribogaine or ECSI#6 could be strengthened. The increased band at a high molecular weight that the authors attribute to the fully glycosylated form is visible. However, this smear, and the band below, look quite different from those in the blot shown in the El-Kasaby et al. reference, raising concerns that the band could be aggregated or dimerized protein rather than a glycosylated form. This concern could easily be addressed by control experiments with appropriate glycosidases, as shown in the reference.

We understand that the appearance of the mature glycosylated species is being criticized, at least in part, because it differs from sharper bands, which can be found in our previously published papers. We stress that the resolution very much depends on the electrophoretic conditions. We addressed the reviewers’ criticism by carrying out the recommended deglycosylation experiments: a representative experiment is shown in (the new) panel F of Figure 5, with lysates prepared from HEK293 cells expressing wild type SERT, from untransfected HEK293 cells and from HEK293 cells, which had been preincubated with 30 µM cocaine, 100 µM ECSI#6 and 30 µM noribogaine. The experiment confirms the band assignment with the upper band(s) M representing the mature glycostylated species (which are resistant to deglycosylation by endoglycosidase H) and the lower band C corresponding to the coregylcoylated species (which are susceptible to cleavage that (as expected) the mature band show a representative degylcosylation by endoglycosidase H). We also think that the immunoblot in panel F ought to satisfy the aesthetic criticism: the bands are sharper/less smeared.

The description of panel F can be found on p. 18, starting in line 7 from bottom to end of page, and reads: “We confirmed the band assignment by enzymatic deglycosylation (Figure 5F): the upper bands (labeled M), which appeared in cells incubated in the presence of ECSI#6 and of norbogaine, were resistant to deglycosylation by endoglycosidase H (which cannot cleave mature glycans). In contrast, the core-glycosylated species (labeled C), was susceptible to cleavage by endoglycosidase H resulting in the appearance of the deglycosylated band D.”

Reviewer #1 (Recommendations for the authors):Figure 1B. The fit for 20 μm 5-HT should be dotted lines. Dashes are shown.

We thank the reviewer for pointing this out. This was corrected in the revised manuscript.

Line 296: the explanation of why uptake leads to accumulation of the K^+^-bound intermediate could be improved for the benefit of a non-expert reader, especially since this relates to a central conclusion of the work.

The return of the K^+^ bound transporter from the inward to the outward facing conformation was shown to represent the slowest reaction (i.e., rate-limiting step) in the transport cycle of SERT is. SERT accumulates in the K^+^ bound intermediates, because these are the reactant states of this reaction. Accordingly, in the revised manuscript we added the following explanation (on p. 9, line 2 to 4):

“(we emphasize that accumulation in the K^+^ bound intermediates is a consequence of the slow return of the K^+^ bound transporters from the inward to the outward facing conformation, which is the rate-limiting step in the transport cycle).”

Figure 3G, H, I y-axis should be labeled as Vmax(units), just as those above are labeled "Km(unit)".

In the revised manuscript we changed the labels of these figures accordingly.

Line 561. It should be explicitly mentioned that To is the outward-facing apo state.

We now mentioned this in the manuscript. The pertinent sentence now has this information inserted and reads, on p. 23, line 5 of second paragraph:

“In the model, the transport cycle begins with the binding of sodium (Na^+^) and substrate (S) to SERT to the apo outward facing state (To).”

Line 586-590. This point is confusingly written and I didn't understand. It should be clarified.

We reworded the sentences (on p. 23, last line and p. 24, lines 1 to 4) to read:

“In contrast, it was only possible to emulate the experimental data by assuming that the binding affinities to the other preferred states (i.e., TiS, TiS, TiNaS and ToNaS, cf. Figure 7F) were higher, that is, in the submicromolar range for TiS, TiS, TiNaS or 1 µM for ToNaS (Figure 7G). Thus, preferential binding of ECSI#6 to these states (rather than TiK) allows for a possible, but highly unlikely solution.”

We hope that this wording is clear.

Reviewer #2 (Recommendations for the authors):- I would like to have access to the raw data of the experiments.

We uploaded the data onto a data deposit (i.e., Dryad).

- The authors use a lot of text to explain their findings and interpretation, which I like. Nonetheless, in places, I find the paper hard to read, which may be caused by the choice of wording. For instance: line 296, "accumulation",

An additional explanation, why SERT accumulates in the K^+^-bound inward facing state has been provided (on p. 9, line 2 to 4):

“(we emphasize that accumulation in the K^+^ bound intermediates is a consequence of the slow return of the K^+^ bound transporters from the inward to the outward facing conformation, which is the rate-limiting step in the transport cycle).”

Line 309-130, "accordingly".…appreciable binding can be measured,

An additional explanation, why SERT accumulates in the K^+^-bound inward facing state has been provided (on p. 9, line 2 to 4):

“(we emphasize that accumulation in the K^+^ bound intermediates is a consequence of the slow return of the K^+^ bound transporters from the inward to the outward facing conformation, which is the rate-limiting step in the transport cycle).”

line 395: it is unclear how continuous cycling would lead to a current for an electroneutral transporter. It would help to explicitly mention that the current is uncoupled.

We have reworded the sentences to explicitly mention the uncoupled current and to provide a more detailed explanation (on p. 14, lines 5 to 11 of the second paragraph): “When challenged with a substrate like 5-HT or amphetamines, SERT carries inward currents. These are comprised of an initial transient peak current and a steady current. The peak current reflects the initial synchronized movement of substrate and co-substrate through the membrane electric field. The steady current arises from an uncoupled current; the underlying channel mode is visited from the K^+^-bound inward-facing state. Thus the steady current reflects the continuous cycling of the transporter in the forward transport mode [24,25].”

Reviewer #3 (Recommendations for the authors):Figure 1A: Should the pKa of these inhibitors be given?Whenever the term "affinity" is used (except in binding studies), it probably should be replaced by "apparent affinity".

We do not see any advantage in reporting pKa (rather than non-logarithmically transformed data); we agree with the reviewer that the term apparent affinities is preferable, when inhibition of the transport cycle is being referred to (appropriate changes were made in the text as recommended). – make sure if apparent affinity was used as mandated.

Figure 2: It would be useful if panels A-C and D-F would have the same x-axis scale, so the reader can more easily compare the IC50 values in the absence and presence of K^+^.

We have redrawn all graphs (including the uptake inhibition curves in Figure 1) such that the xaxes have an identical scale (and an axis-break to show the control values at 0 inhibitor).

Line 302: For the reader unfamiliar with SERT pharmacology, it should be explained why citalopram was used for the binding inhibition experiments.

We provide an explanation, why we used tritiated citalopram as a radioligand (on p. 9, lines 6 to 10):

“We resorted to radioligand binding experiments with [^3^H]citalopram to address which of the two possibilities can explain the experimental observations: [^3^H]citalopram is a radiolabeled high-affinity inhibitor of SERT, which binds to the outward facing conformation.”

Figure 3 D-F: The plots should be made in a regular x-y fashion, with a linear x (concentration) axis scale. That would make it easier to see if the cocaine data adhere to the regular competitive enzyme inhibition mechanism equation, in which Km is expected to increase in a linear fashion with inhibitor concentration. My feeling is that the slope of that curve is too shallow for cocaine to account for the full competitive mechanism.

An inset was added to Figure 3 D-F; the inset to Figure 3D shows that the (apparent) KM of serotonin increases in (reasonably) linear manner with increasing cocaine concentrations.

Line 393: It is evident from the data that cocaine and noribogaine induce a peak current, but ECSI#6 does not. However, the authors do not explain why this is the case. This seems to be an important detail that could further inform on the inhibition mechanism.

We now addressed this in the text. We provide two alternative explanations for why ECSI#6 failed to produce a peak current (on. p. 13, lines 14 to 21 of the second paragraph). “In the presence of physiological ion gradients and in the absence of substrate, SERT dwells primarily in the outward-facing conformation. A precondition for a ligand to elicit a peak current is, therefore, the ability to bind to outward facing states. The failure of ECSI#6 to produce a peak current is, thus, consistent with the interpretation that this compound binds exclusively to inward facing states. It should be noted, however, that cocaine and noribogaine carry a charge while ECSI#6 does not. Thus, it is not possible to formally rule out the alternative explanation that ECSI#6 fails to generate a peak current, because it is uncharged.”

Figure 4 D-E: Linear kapp vs concentration relationships are typically observed in situations where ligand binding is rate limiting, often under diffusion control. From the slope of the relationship, the bimolecular rate constant for binding can be obtained. What is this value? I assume that it is far below a diffusion-controlled rate constant?

We conducted the requested analysis. We inserted the following text starting on line 8 to the end of the first paragraph on p. 14:

“We obtained estimates for the k_on_ and k_off_ of noribogaine and ECSI#6, respectively, from the slopes and the y-intercepts of the linear regression in panels D and E. The extracted k_on_ and k_off_ values were 1.7*10^5^±1.6*10^4^M^-1^*s^-1^ and 1.4±0.3 s^-1^ and 2.1*10^4^±3.5*10^3^M^-1^*s^-1^ and 1.7±0.5 s^-1^, for noribogaine and for ECSI#6, respectively. However, these rate constants do not allow for estimating a kinetically calculated K_D_. In fact, the rate of the inhibitor-induced current block is comprised of both, (i) the binding rates of the inhibitor and (ii) the rates of the conformational transitions, which SERT undergoes during substrate transport. Thus, the protocol employed (cf. Figure 4B) does not afford an accurate determination of inhibitor binding rates.” We hope that this explanation addresses the question of the reviewer.

Figure 5D: This Western blot is not very convincing. (1) The band at 110 kD is very weak, and this reviewer has trouble reconciling the bands with the data shown in Figure 5E. (2) What is the band at 60 kD? This blot needs further explanation.

See above, reply to point 6 of editorial summary.

Figure 7 legend: No information is given on the microscopic rate constants for inhibitor binding/dissociation. Are these assumed to be in equilibrium with respect to the other transport steps?

We apologize for this oversight. We assumed for all inhibitors the same association rate (i.e., k_on_ = 10^6^ M^-1^*s^-1^) and set the K_Ds_ to the desired values by adjusting the corresponding dissociation rates (K_D_ = k_off_/k_on_). We chose an association rate that is approximately two orders of magnitude lower than the diffusion limit because a rate in this range is typical for SERT inhibitors (see refs. 7 and 23). In the legend of figure 7 we now state:

“In the simulation we assumed that all inhibitors bind to SERT with the same association rate (i.e., k_on_(inhibitor) = 10^6^ M^-1^*s^-1^) and we set the K_D_s to the desired values by adjusting the corresponding dissociation rates (k_off_inhibitor; K_D_ = k_off_/k_on_). A value in the range of 10^6^ M^-1^*s^1^ is typical for the k_on_ of SERT inhibitors [7, 23]”

Figure 8: The plots of simulated vmax vs. inhibitor concentration should also be shown.

We now included the requested graphs in Figure 8 (see new panels E and F). To the legend we added the following description: In (E) and (F) we show the V_max_ values obtained from the simulation as a function of the noribogaine and ECSI#6 concentration, respectively